# De novo motor learning creates structure in neural activity that shapes adaptation

Joanna C. Chang[1], Matthew G. Perich [2,3], Lee E. Miller [4], Juan A. Gallego [1,5] ✉ & Claudia Clopath [1,5] ✉

Animals can quickly adapt learned movements to external perturbations, and their existing motor repertoire likely influences their ease of adaptation. Long-term learning causes lasting changes in neural connectivity, which shapes the activity patterns that can be produced during adaptation. Here, we examined how a neural population's existing activity patterns, acquired through de novo learning, affect subsequent adaptation by modeling motor cortical neural population dynamics with recurrent neural networks. We trained networks on different motor repertoires comprising varying numbers of movements, which they acquired following various learning experiences. Networks with multiple movements had more constrained and robust dynamics, which were associated with more defined neural 'structure'—organization in the available population activity patterns. This structure facilitated adaptation, but only when the changes imposed by the perturbation were congruent with the organization of the inputs and the structure in neural activity acquired during de novo learning. These results highlight trade-offs in skill acquisition and demonstrate how different learning experiences can shape the geometrical properties of neural population activity and subsequent adaptation.

From walking to grasping objects, movement enables us to interact with the world. Mastering a skill requires many hours of practice, be it during development or in adulthood. In contrast to long-term skill learning, adapting existing skills to environmental perturbations is a much faster process: after learning to ride a bike, adapting to foggy weather or uneven roads is much easier. Existing motor repertoires acquired through long-term learning likely form the foundation for short-term motor adaptation, but it is unclear how different repertoires can affect adaptation, even for common experimental perturbations like visuomotor rotations (VR)[1] or force fields[2]. This lack of understanding stems from the experimental challenge of characterizing an animal's entire behavioral repertoire learned throughout their lifetime, and implies that the interplay between available neural activity patterns and rapid adaptation remains largely unknown[3].

Recent work has focused on the coordinated activity of neural populations to begin to shed light on the neural basis of motor adaptation[4–6]. In this neural population view, brain function is not built upon the independent activity of single neurons, but rather on specific patterns of neural co-variation (from now on, simply 'activity patterns')[7–10]. In practice, these activity patterns can be examined by building a neural state space (from here, referred to as 'neural space') where each point denotes the state of the neural population. Numerous studies[11–16] have found that the activity of even hundreds of simultaneously recorded motor cortical neurons is well captured by relatively few population-wide activity patterns, an observation consistent with neural population activity being constrained to a low-dimensional surface—a 'neural manifold'—that can be estimated by applying a dimensionality reduction method[17–21]. Studies comparing neural population activity patterns evoked by sensory stimuli and

[1]Department of Bioengineering, Imperial College London, London, UK. [2]Département de Neurosciences, Faculté de Médecine, Université de Montréal, Montréal, QC, Canada. [3]Mila, Québec Artificial Intelligence Institute, Montréal, QC, Canada. [4]Departments of Physiology, Biomedical Engineering and Physical Medicine and Rehabilitation, Northwestern University and Shirley Ryan Ability Lab, Chicago, IL, USA. [5]These authors jointly supervised this work: Juan A. Gallego, Claudia Clopath. ✉e-mail: jgallego@imperial.ac.uk; c.clopath@imperial.ac.uk

optogenetic stimulation provide direct evidence of neural manifolds reflecting circuit connectivity constraints[22,23]. For example, optogenetic stimulation of a small subset of stimulus-selective neurons evokes the same activity patterns in the manifold of a population of simultaneously recorded neurons as those observed during 'natural' stimulus presentation[22], indicating that neural manifolds are at least partly shaped by the underlying connectivity of the network. This presumed relationship between circuit constraints and neural manifold geometry is further supported by the preservation of neural manifolds in the anterior thalamus[20,24] and medial entorhinal cortex[19] between active behavior and sleep.

Previous work suggests that long-term learning causes changes in circuit connectivity[25–28], which may in turn change the geometrical properties of an existing neural manifold as well as the activity within it —the so-called 'latent dynamics'. In contrast, experimental and modeling studies of motor adaptation suggest that this form of rapid learning may be achieved without any changes in motor cortical connectivity or, if present, these changes would be very small and low-dimensional[6,29]. In good agreement with these findings, using a brain-computer interface that mapped motor cortical activity onto computer cursor movements, Sadtler et al. showed that it is easier to adapt to perturbations that require only neural states that lie within the existing neural manifold: while animals can learn to produce activity patterns within the existing manifold in a matter of minutes or hours[16], producing activity patterns outside the existing manifold takes several days[30]. To learn outside-manifold perturbations, new activity patterns need to be used, which may necessitate changing the synaptic connectivity of the circuit[30,31]. Combined, these results suggest that long-term motor learning changes the circuit connectivity to create new neural population latent dynamics, whereas short-term motor adaptation may reuse existing ones[32].

Yet, studying the interplay between an animal's existing motor repertoire and its ability to adapt a known behavior remains challenging due to the complex organization of existing motor skills within neural space. Similar behaviors (e.g., various wrist manipulations or grasping tasks) may share similarly oriented task-specific neural manifolds[13], while dissimilar behaviors (e.g., reaching and walking in mice) may have almost orthogonal manifolds even if they share similar movements[15]. Furthermore, experimentally, it is impossible to quantify the entire repertoire of motor skills that an animal has, which poses a challenge to investigate how its behavioral repertoire influences motor adaptation. Similarly, experimenters also often lack access to how skills were acquired in the first place, which poses an additional challenge to understanding the neural basis of learning because the de novo learning process influences future learning[33,34], e.g., based on whether the training examples are meaningfully organized with respect to each other[35]. Finally, in virtually all motor adaptation studies, animals must adjust to perturbations on a specific laboratory task they have already learned, and adaptation is only examined with respect to a 'baseline period'[1,2,4–6,16,36,37]; this approach ignores, for practical reasons, the relationship between the adapted behavior and all the other motor skills the animal has previously acquired, as well as how these skills were initially learned.

To overcome the experimental challenges of assessing an animal's lifelong experience, here we examined how the existing motor repertoire can affect adaptation differently using recurrent neural networks (RNNs). Similar networks have been able to reproduce motor output and key features of latent dynamics from experimental recordings[38–41], including during adaptation[29,31]. We modeled the latent dynamics of the motor cortex during de novo learning and subsequent adaptation. We trained our networks on different repertoires with varying numbers of movements through a variety of learning experiences using movement trajectories modified from experimental recordings of monkey reaches[6,42]. We hypothesized that networks with larger motor repertoires would adapt to perturbations more easily since they are already able to produce a broader set of activity patterns, but only if perturbations require changes that are 'meaningful' with respect to their learning experience.

By systematically training networks in two stages comprising of de novo learning and subsequent adaptation, we found that larger repertoire networks could adapt to perturbations more quickly, but only under certain circumstances. The way the latent dynamics of multiple movements were organized in neural space shaped subsequent adaptation: adaptation was facilitated when the latent dynamics were organized in a way that was congruent with the changes required by the perturbation, and when only small changes in motor output were needed. This suggests that ease of adaptation is affected not only by its relation to the existing manifold[16,30], but also by the organization of the latent dynamics within it. Crucially, this organization is affected by past learning experiences. These observations highlight an inherent trade-off in skill acquisition: mastery of more movements better defines the structure of the neural manifold. This, in turn, facilitates adaptation that requires small changes in behavior, but harms adaptation that requires large changes or when the learning experience is not directly relevant to the new behavior.

## Results

### Probing the impact of de novo learning on subsequent adaptation with RNNs

To understand how a neural population's existing activity patterns affect its ability to change its activity, we used RNNs to model motor cortical neural population dynamics following de novo learning and subsequent adaptation (Fig. 1a).

To model de novo learning, we trained the networks to perform different repertoires with different numbers of movement directions. The movements were modified from experimentally recorded center-out reaches from monkeys (data from ref. 6, Fig. 1b, Methods). For each reach, monkeys were first presented with a visual target; after a

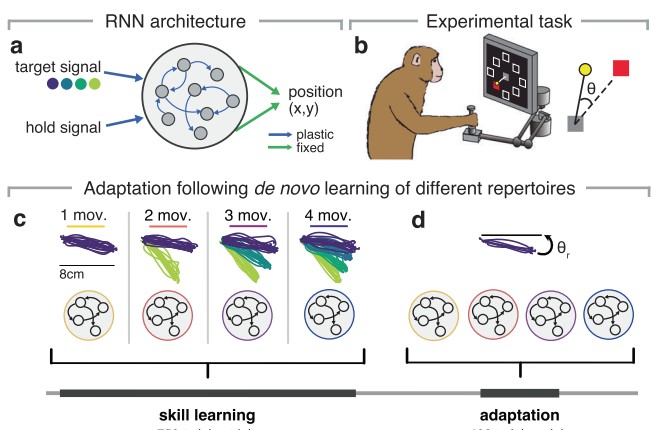

**Fig. 1 | Probing de novo learning and adaptation with RNNs. a.** RNNs were trained to produce 'hand positions' during a standard center-out reaching task as output. They were given a target signal that specifies the reach direction, and a hold signal that indicated movement initiation. The target signal indicated the angular direction, unless otherwise specified. The input and recurrent weights were learned (plastic), while the output weights were fixed, unless otherwise specified. **b** RNNs were trained on hand trajectories modified from experimental recordings of monkeys performing a similar center-out reaching task (left). Subsequent adaptation was studied using a classic visuomotor rotation paradigm (VR, right), in which visual feedback is rotated by a fixed angle around the center of the workspace. **c** Networks were trained on repertoires with different numbers of movements (from one to four) to model de novo learning. All multi-movement networks (2 mov., 3 mov., and 4 mov.) covered the same angular range. **d** Networks were later trained to counteract VR perturbations for only one common movement to understand the influence of existing motor repertoires on adaptation.

variable delay period, a 'go' cue indicated they could execute the movement. To address our first hypothesis that having a larger motor repertoire would facilitate adaptation, we used repertoires of different sizes, ranging from one to four movement directions (Fig. 1c). Single-movement repertoires comprised a single reach to −10° while multi-movement repertoires comprised movements in directions equally spaced between −10° and −50°, unless otherwise specified. Note that we intentionally limited our networks to four movements for simplicity to make our simulation results interpretable. Importantly, all networks learned to produce the −10° direction, allowing for comparisons across networks trained on different repertoires. We further kept all training parameters constant to simulate a common de novo learning experience. Networks were given the angular direction of these movements as inputs along with a go cue (occurring 0.5–1.5 s after the direction cue) to mimic the instructed delay-reaching task performed by the monkeys (Methods).

To model motor adaptation, we subsequently trained these networks to counteract a visuomotor rotation (VR) on the shared movement direction they had all learned (Fig. 1d). VR is a commonly used experimental paradigm to examine motor adaptation[1,5,6,43–45] in which a rotational transformation is applied to the motor output. In this case, we applied a 10° rotation (counterclockwise), which the networks had to

counteract by producing output in the opposite direction. By probing adaptation on only one common movement, we could assess how entire motor repertoires contribute to the adaptation of a given movement, and compare the adaptation performance across repertoires.

## Multiple movements produce more constrained and robust dynamics that are structured in neural space

We first trained networks to produce repertoires comprising one to four different movements to understand resulting differences in the underlying network activity. Following the initial de novo learning phase, all the networks were able to learn each of the repertoires (Fig. 2a) with comparable performance (Fig. 2b), as quantified by the mean-squared error between target movement trajectories and produced movement trajectories. This indicates that any differences in their ability to adapt will not be due to how accurately they can generate motor output, but rather to differences in the network dynamics that can be produced based on their acquired motor repertoires.

To compare across different repertoires, we examined the neural dynamics as the networks were producing the common −10° movement. We focused on network activity during both preparation and execution of the same target motor output. We calculated

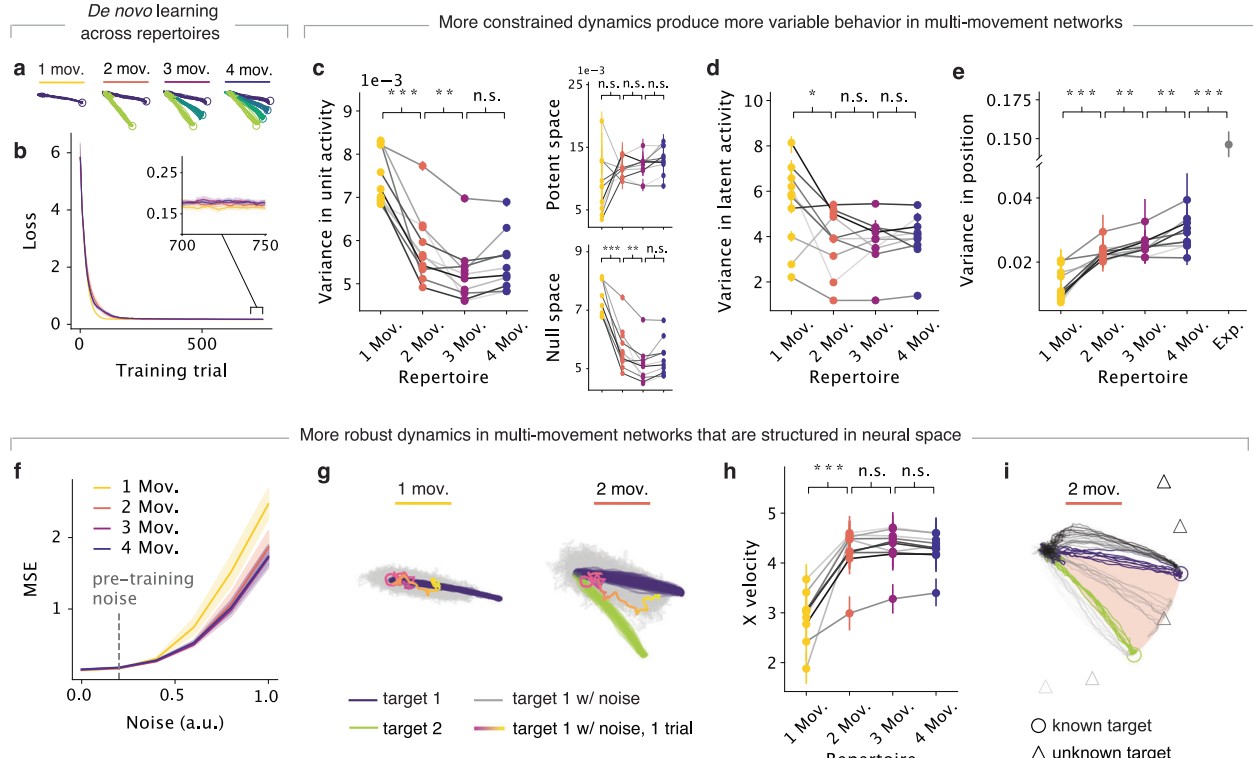

**Fig. 2 | Networks that have acquired multiple movements produce more constrained and robust dynamics that are structured in neural space. a** Motor output of networks trained on different repertoires following de novo learning. **b** Loss during initial de novo learning. Loss was calculated as the mean-squared error between the network output and target positions. Line and shaded surfaces, smoothed mean, and 95% confidence interval across networks of different seeds (*n* = 10 random seeds). **c–f**, **h** Following de novo learning, networks were tested on the one shared movement. **c** Variance in unit activity for networks trained on different repertoires. Variance was calculated for each unit for each time step across trials. Inset: Variance in the output-null subspace and output-potent subspace of the unit activity with respect to the produced output. Individual lines, different random seeds (*n* = 10); circles and error bars, median and 95% confidence intervals with bootstrapping. *** denotes 0.001, ** 0.01, * 0.05 for one-sided Wilcoxon signed-rank tests. **d** Same as panel (**c**), but

for the variance in latent dynamics. **e** Same as panel **c** but for variance in motor output. Variance for single reaches for monkeys trained on the center-out reach task (data from ref. 6) is included for comparison. Note that the networks, which only know a few movements, generally have less variance than monkeys. **f** Mean-squared error of output when noise of increasing magnitude is added to the neural activity. Line, shaded area, median, and 95% confidence interval. **g** Motor output with (gray, pink gradient) and without (purple, green) increased noise added (η = 1) for example networks with one- and two-movement repertoires. Color gradient, time course of the movement execution (legend). **h** Same as panels **c–e** but for velocity in the *x* direction for motor output when noise (η = 1) was added as per panel (**g**). **i** Motor output for different target cues for a sample network with two-movement repertoire. Colors, different target cues; circles and triangles, target endpoints for each movement; pink background, range of known movements.

the population latent dynamics by projecting the activity of all 300 units in our networks onto a lower-dimensional neural manifold identified by performing principal component analysis[7,17] (PCA) (Methods). The main differences in the activity produced by networks with different repertoires was between single-movement and multi-movement (two, three, or four movements) networks: multi-movement networks had less variance across trials to the shared target in both single unit activity (Fig. 2c left, $P = 9.8 \cdot 10^{-4}$, Wilcoxon signed-rank test; Supplementary Fig. 1a–c; Methods) and latent dynamics (Fig. 2d, $P = 0.02$, Wilcoxon signed-rank test; Supplementary Fig. 1d–f) than single-movement networks, suggesting that the network activity becomes more constrained when multiple movements need to be embedded in the neural space. Intriguingly, despite their lower variance in activity, multi-movement networks had greater variance in motor output (Fig. 2e, $P = 9.8 \cdot 10^{-4}$, Wilcoxon signed-rank test; Supplementary Figs. 1g–i, 2)—which was more comparable to experimental reaches (Fig. 2e)—even though all the networks were performing the same movement. How can these apparently contradictory observations be reconciled? We hypothesized that the more variable latent dynamics of single-movement networks may lie in directions of activity space that do not affect motor output, i.e., those defining the 'output-null' subspace[46]. Separately computing the variance of the latent dynamics within the output-null subspace and the 'output-potent' subspace (i.e., the directions that affect the network output), confirmed this prediction: the greater variability of the latent dynamics of single-movement networks was largely confined to the output-null subspace, and thus did not lead to more variable motor output (Fig. 2c right; Methods). These results also held when networks were trained on synthetic reaches that were more stereotyped than actual monkey reaches (Supplementary Fig. 3) and across different ways of calculating inter-trial variability (Supplementary Fig. 3; Methods). Thus, greater constraints in the network dynamics were characteristic of multi-movement networks, but these more constrained dynamics did not necessarily translate into more consistent behavioral output.

How may these constraints affect the ability of multi-movement networks to generate robust latent dynamics driving motor output? We predicted that they may lead to greater robustness against noise[47]. We examined this by changing the amount of simulated noise applied to the unit activity, and saw that multi-movement networks were indeed more robust against higher noise levels than single-movement networks (Fig. 2f and Supplementary Fig. 4g). To probe this further, we increased the noise five-fold compared to pre-training and measured how the output was affected (examples in Fig. 2g and Supplementary Fig. 4a, b, d, e, with the two-movement network representative of multi-movement networks). Intriguingly, while motor output for single-movement networks circled back and became twisted, that for two-movement networks shifted toward previously learned outputs at this high noise level (Fig. 2g and Supplementary Fig. 4e). This systematic shift indicates that learning multiple movements creates structure in the neural space that maps to structure in the motor output; that is, the trajectories described by the latent dynamics driving each movement are organized in neural space in a way that is congruent with that of the movements. With this underlying 'structure', noise in the activity space caused two-movement networks to explore other activity states that led to movements intermediate to those that had been previously learned (Fig. 2g and Supplementary Fig. 4b, e). This led to mostly linear output trajectories with greater forward movement, as quantified by the velocity in the $x$ direction during execution (Fig. 2h, $P = 9.8 \cdot 10^{-4}$, Wilcoxon signed-rank test), in contrast to the twisted and tangled movements produced by single-movement networks—with these differences in motor output being appropriately reflected in neural space (Supplementary Fig. 4c, f). Moreover, with their underlying structure, two-movement networks were able to generate intermediate movements

that they had not previously learned when probed with the appropriate input signal (Fig. 2i and Supplementary Fig. 5b, c). Combined, these results suggest that learning multiple movements creates structure in the neural space that effectively makes more intermediate activity patterns available. The availability of these additional activity patterns may facilitate adaptation.

## Networks with larger repertoires can adapt to a perturbation more easily

To directly assess our first prediction that the additional structure in the neural space of multi-movement networks may facilitate adaptation, we applied the same small VR perturbation of 10° to networks with different movement repertoires (Fig. 3a, b). In general, networks that had learned larger repertoires were able to adapt more quickly (Fig. 3b, c and Supplementary Fig. 5d, e), and without catastrophic forgetting (Supplementary Fig. 6). Comparison of adaptation performance across networks with different motor repertoires revealed two trends. First, all three types of multi-movement networks adapted much more rapidly than single-movement networks, as predicted. Within 100 adaptation training trials, multi-movement networks converged to a performance comparable to baseline. Single-movement networks, in contrast, could not fully adapt and had average errors 20% larger than that of the multi-movement networks (Fig. 3b). Second, within multi-movement networks, those with larger repertoires also adapted more quickly than those with smaller repertoires (Fig. 3c). Thus, having larger motor repertoires, which was associated with greater structure in activity space following the initial de novo learning process (Fig. 2), facilitated adaptation.

## The initial de novo learning experience shapes the structure of neural activity and influences subsequent adaptation

The previous simulations support our first hypothesis that having a larger behavioral repertoire facilitates adaptation (Fig. 3), but is more always necessarily better? Task performance is shaped by how behavioral components are acquired[33,34], including by whether the examples presented during the initial learning process are meaningfully organized with respect to each other[35]. This led us to hypothesize that the de novo learning experience in and of itself would shape adaptation.

To test this hypothesis directly, we trained a second series of networks to perform the exact same reaching task following a different learning experience. We modeled this different learning experience by altering the encoding of the input cues, predicting that this

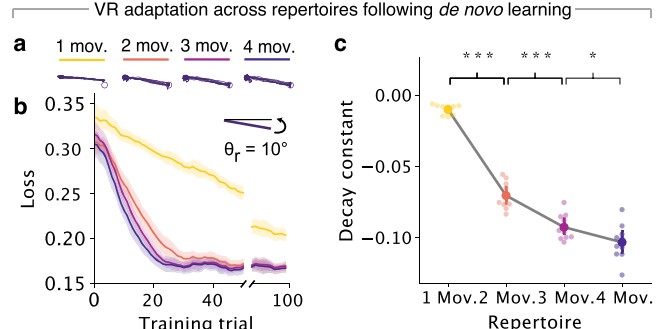

**Fig. 3 | Networks with larger repertoires can adapt to perturbations more easily. a** Motor output of networks trained on different repertories (legend) following adaptation to a counterclockwise VR perturbation. **b** Loss during adaptation, calculated as the mean-squared error between the network output and target positions. Line, shaded surfaces, smoothed mean, and 95% confidence interval across networks of different seeds. **c** Decay constants for exponential curves fitted to the loss curves in panel (**b**). Circles and error bars, means and 95% confidence intervals with bootstrapping. *** denotes 0.001, ** 0.01, * 0.05 for two-sided paired $t$-tests. Data includes $n = 10$ networks from ten random seeds for each repertoire.

would alter the structure of the network activity. Thus far, we have used continuous angular inputs that specified the direction of the targets (Fig. 4a). This mimics the process of learning to reach to an endpoint location based on a cue that indicates the take off angle of the movement. An alternative way to perform the same task would be to learn to produce each reach in response to an arbitrary cue, in which case there is no meaningful relationship between target-specific cues. We implemented this alternative learning experience by training networks to produce the same motor output using categorical, one-hot encoded binary vectors that had no angular information (Fig. 4d, Methods). Even after matching performance following de novo learning (Fig. 4h, i top), the latent dynamics of these networks with categorical inputs no longer had the same organization during preparation as the motor output (Fig. 4e), as was the case for the networks with continuous angular inputs Fig. 4b). This confirms that we could indeed enforce a different structure in neural space by changing the initial learning experience.

To further characterize the structure in the activity space of each network and compare it across networks with different learning experiences (angular vs. categorical inputs), we measured the Euclidean distances in the neural manifold between the latent trajectories at corresponding time points for different movements (Methods, Fig. 4c, f right). To allow for comparison between different networks that have different neural spaces, we normalized these distances between movements by the distances between adjacent time points along the same latent trajectory, which should be comparable across networks. We first focused on the networks with angular inputs. During both the preparation and execution epochs, the latent trajectories of these networks were organized similarly to the movements themselves, with movements reaching adjacent targets also adjacent in neural space (Fig. 4a, b and Supplementary Fig. 7). In contrast, for the networks with categorical inputs, the latent dynamics during preparation for different movements no longer had the same organization as the motor output (Fig. 4d, e). Thus, the organization of the latent dynamics in both classes of networks reflected the structure of the inputs—quantified by calculating the cosine dissimilarity between pairs of input vectors (Methods)—suggesting that the structure of the inputs imposes a congruent structure in neural space (Fig. 4g, Methods). This was the case even if the motor output was extremely similar across network classes (Fig. 4h, i) after the initial de novo learning phase.

Having two classes of networks that perform the initial center-out task equally well following different learning experiences allowed us to directly test the influence of the de novo learning experience in subsequent adaptation. For the angular input networks, the activity in neural space was congruently structured (Fig. 4b), with the latent trajectories organized by the angular direction of the movements. Since a VR perturbation requires angular changes on the motor output, we predicted that the greater congruence between the structure in the neural space of angular input networks and the angular changes required by the perturbation would facilitate adaptation. Indeed, while the performance for networks with angular and categorical inputs was comparable following adaptation (Fig. 4h, i bottom), adaptation was, in general, faster for networks with more congruence (angular input networks) than networks with less congruence (categorical input networks) (Fig. 4j), confirming that the experience during initial de novo learning influences adaptation. Moreover, for the angular input networks, adaptation was also faster for those with larger movement repertoires (Fig. 4h), but this trend was absent for the categorical input networks (Fig. 4i). Single-movement networks lacking structure in neural space were unaffected by the difference in input encoding (Fig. 4h, i). These results support our second hypothesis that it is not only the motor repertoire that shapes learning: how movements are initially acquired determines the organization of

neural activity which also shapes further learning. In particular, the congruence between the structure of the latent trajectories in neural space and the structure in the perturbation is important to facilitate adaptation. Interestingly, when we applied this measure of congruence to experimental data of monkeys performing a center-out reaching task (Methods), we could predict the rate of adaptation to a VR perturbation based on the degree of congruence (Supplementary Fig. 8f). This lends experimental support for a potential role of congruence between the structure of latent trajectories and that of the perturbation in adaptation.

## Adaptation is facilitated by exploiting intermediate states within structured activity

Having established that the motor repertoire and the learning experience jointly shape adaptation, we sought to understand what property of the angular input networks allowed networks with greater repertoires to adapt faster. We had previously shown that the structure of these multi-movement networks organizes the neural dynamics and allows them to produce untrained, intermediate movements (Fig. 2g, i and Supplementary Fig. 5b, c). Thus, a larger movement repertoire in angular input networks may lead to a more defined neural structure that allows the networks to produce more intermediate activity states. These intermediate states may facilitate adaptation by reducing the need to learn new activity patterns. When modeling learning using RNN models, connectivity must be altered in order to change the activity patterns that can be produced[3,29]. This implies that if large repertoire networks do exploit intermediate states provided by additional movements, they would require smaller adaptive weight changes than smaller repertoire networks. This was indeed the case, but, as anticipated from their differences in the time course of learning (Fig. 4h, i), it only occurred in the more congruently structured networks with angular inputs, not in the networks with categorical inputs (Fig. 4k).

How do intermediate states contribute to the faster adaptation of networks with angular inputs and larger motor repertoires? To explore the role of these intermediate states, we examined how the network activity evolved during adaptation. If we assume that intermediate states exist between the latent trajectories for each movement, we would expect the latent trajectories to move along these states during adaptation, in the direction of other existing latent trajectories that are in the direction of the desired motor output. To measure this, we defined a 'deviation angle' that quantifies how changes in the trajectories during adaptation deviate from the path afforded by the existing potential intermediate states created during de novo learning (Fig. 4l). To validate this metric, we calculated deviation angles for motor cortical recordings from monkeys performing the same VR task[6] and saw they were comparable to those calculated for our networks (Supplementary Fig. 8). If the network is using existing intermediate states, we would expect these angles to be small. Indeed, networks with angular inputs that had more congruent structure than the networks with categorical inputs also had smaller deviation angles (Fig. 4m), with the differences paralleling those seen in the rate of adaptation (loss curve decay constants) and relative weight changes (Fig. 4j, k). Thus, how latent activity is structured in neural space has a large impact on motor adaptation. This structure is jointly shaped by the motor repertoire and how the latent dynamics underlying these movements are organized, which is determined by the learning experience, here modeled as the type of cues. Adaptation may only be facilitated if this structure is congruent to the changes on the motor output that are required by the perturbation (Fig. 4h, i). Without any structure to guide adaptation, single-movement networks with both types of inputs adapted the most slowly to our VR perturbation. Without congruent structure, multi-movement networks with categorical inputs adapted more slowly than those with angular inputs. Finally, with more movements to provide more

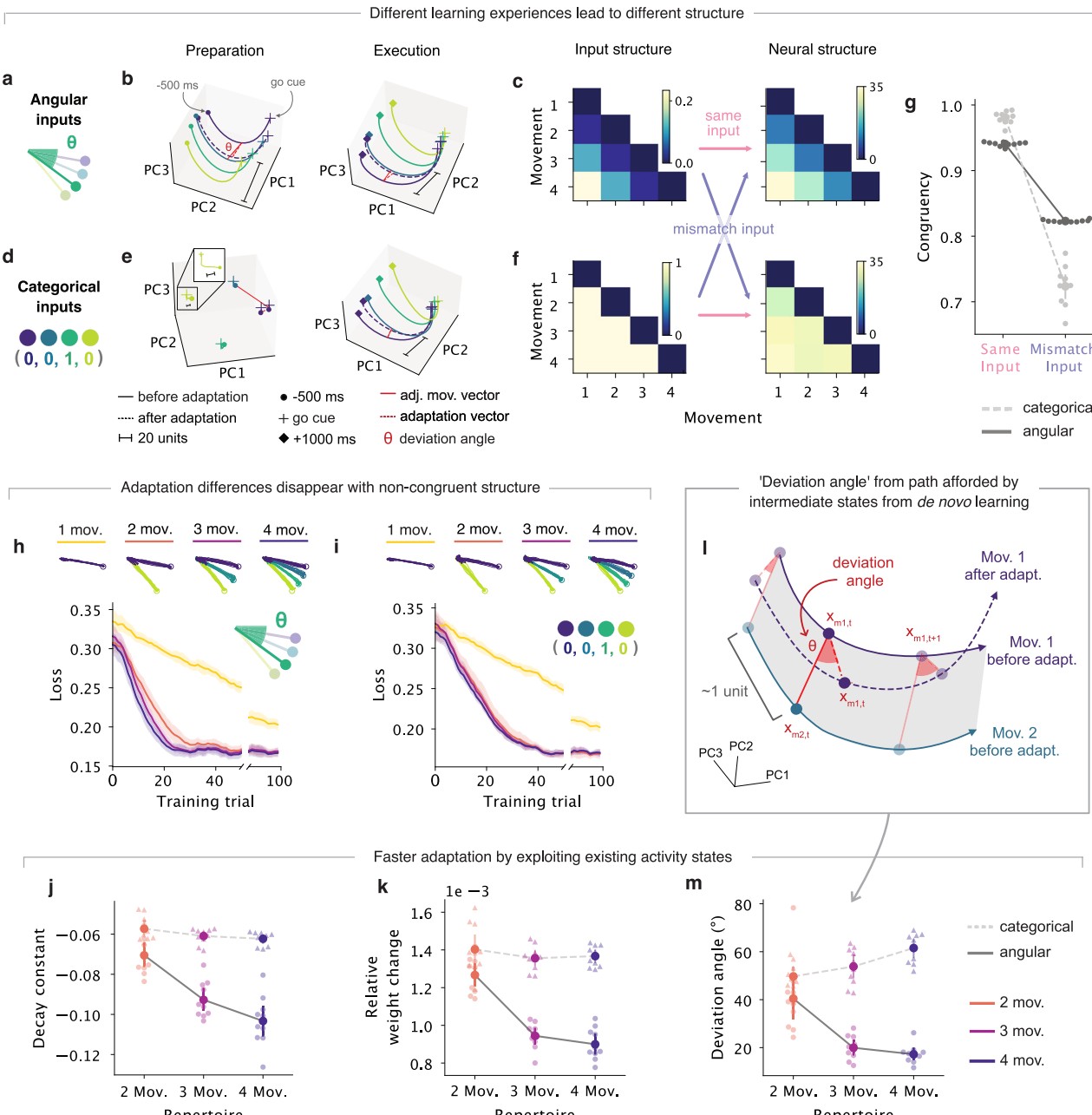

**Fig. 4 | The structure in neural space of multi-movement networks resulting from de novo learning is responsible for their patterns in adaptation.** Networks were given either angular (**a**) or categorical (**d**) inputs to simulate different learning experiences. After training on different repertoires, they had to adapt to a 10° VR perturbation. **b** Latent activity, for example, network with angular inputs trained on four movements during preparation (500 ms before go cue) and execution (1000 ms after go cue). Each trace corresponds to the trial-averaged activity for each movement projected on the neural manifold computed before adaptation. Solid lines, activity before adaptation; dotted lines, activity after adaptation. **c** Left: Input structure, measured as the cosine dissimilarity between input vectors for pairs of movements for the network in Panel b. Right: Neural structure, measured as the normalized median Euclidean distances between latent trajectories during preparation and execution for different movements for the same network. **e**, **f**. Same as panels (**b**, **c**) but for a categorical input network. Inset in Panel **e**: zoomed view. **g** Congruency between the input and neural structure, quantified as the

Pearson's correlation between their dissimilarity matrices. Congruency for mismatched input-activity pairings shown as control. Circle and error bars, mean of congruency values for each seed ($n = 10$) and 95% confidence intervals (CIs) with bootstrapping. **h** Motor output following skill-learning for angular input networks. Bottom: Loss during adaptation training. Traces, shaded surfaces, smoothed mean, and 95% CIs across networks of different seeds ($n = 10$). **i** Same as panel **h** but for categorical input networks. **j** Decay constants for exponential curves fitted to the loss curves in panels (**h**, **i**). Circles, error bars, means, and 95% CIs with bootstrapping. **k** Relative weight changes during adaptation. Circles and error bars, means of the median changes across all weights for each seed, and 95% CIs with bootstrapping. **l** Schematic for a 'deviation angle' between the 'adjacent movement vector' (red solid line in panel **b**, **e**) and the 'adaptation vector' (red dotted line in panel **b**, **e**). **m** Deviation angles during adaptation. Circles and error bars, means of the median deviation angles across all time steps for each seed and 95% CIs with bootstrapping.

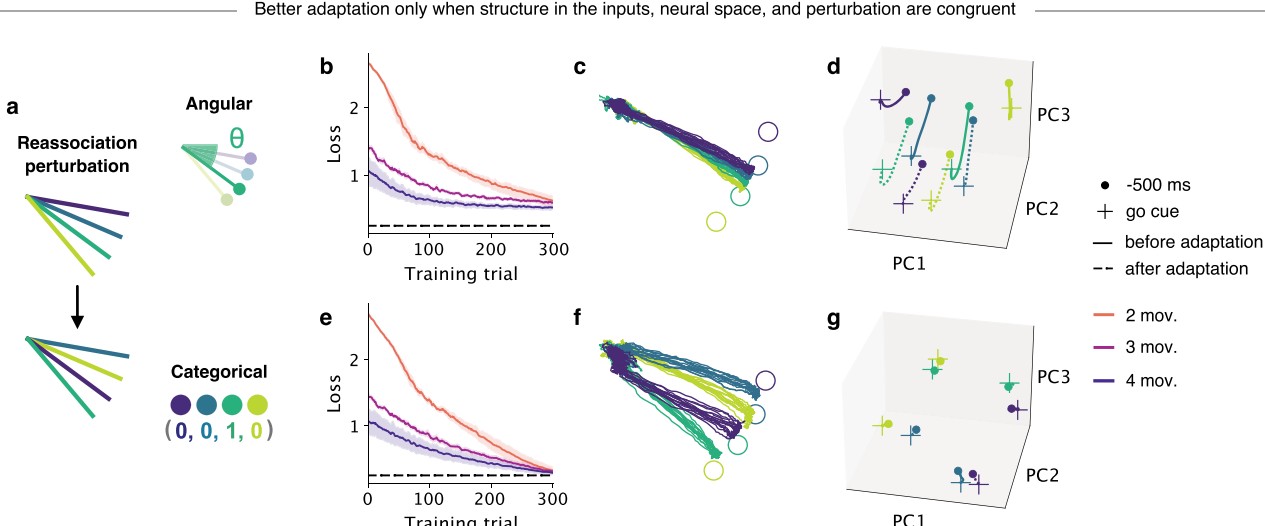

**Fig. 5 | The structure in neural space arising from de novo learning can facilitate or hinder adaptation. a** Networks with angular or categorical inputs adapted to a reassociation rather than a visuomotor rotation perturbation. **b** Loss during adaptation training for networks with angular inputs with varying motor repertoires. Traces and shaded surfaces, smoothed mean and 95% confidence intervals across networks of different seeds (*n* = 10). The dashed line indicates a loss of 0.25 and is included for easier comparison between the loss in networks with angular or categorical inputs. **c** Motor output following adaptation training. **d** Latent trajectories during preparation (see Fig. 4a) for one example seed. **e**–**g** Same as panels (**b**–**d**) but for networks with categorical inputs. Note that networks with angular encoded inputs could not adapt to the perturbation, whereas networks with categorical inputs did since their structure was congruent with the changes required by the perturbation.

intermediate states across the structure, networks with the largest repertoires adapted the fastest.

## Structure in neural space resulting from the learning experience can facilitate or impede adaptation

Our previous results suggest that the structure across the inputs, the neural space, and the changes in the output required by the perturbation all need to be congruent for adaptation to be facilitated. To test this directly, we examined how networks adapted to a different type of perturbation that was congruent with the organization of categorical inputs but not with that of angular inputs, as was the case for the VR perturbation. In this 'reassociation' perturbation (Fig. 5a), the cues and targets were rearranged such that the network needed to re-associate learned reaches to different known target cues[29]. Since this perturbation requires adaptation to categorical rather than angular changes, we predicted that multi-movement networks with categorical inputs would adapt more easily than those with angular inputs.

Indeed, networks with categorical inputs adapted to the reassociation perturbation (Fig. 5e, f), whereas networks with angular inputs were unable to adapt (Fig. 5b, c and Supplementary Fig. 9) despite comparable performance following de novo learning (Fig. 4h, i top). For both classes of networks, the latent trajectories maintained their general structure (Fig. 5d, g), suggesting that the networks adapted by reusing activity patterns in the existing intermediate states rather than by exploring new ones. However, for networks with angular inputs, neighboring latent trajectories interfered with one another during adaptation such that the adapted motor output became overlapped (Fig. 5c, d). Thus, the existing structure in neural space now harmed adaptation instead of facilitating it. Together, these results further support a fundamental relationship between the structure of neural activity following de novo learning and the ability to adapt to a subsequent perturbation: the underlying neural structure can facilitate adaptation if it is congruent to the changes required by the perturbation, but it can also hinder adaptation if the structure of the perturbation is incongruent. Having a well-defined structure in activity space also hindered adaptation to other types of perturbations that required greater changes in the motor output (Supplementary Fig. 10). Thus, structure in the neural space can shape adaptation by facilitating

adaptation under small changes or interfering with adaptation under larger changes. In conclusion, different de novo learning experiences can lead to differences in the organization of neural activity, which can, in turn, lead to dramatic differences in adaptation even across networks with the same motor repertoire.

## Generalization to more complex motor repertoires

Our results so far have focused on networks producing reaching movements. To examine the generalizability of our findings to a more complicated task, we trained networks to produce up to four different elliptical movements by generating cosinusoidal and sinusoidal outputs with various amplitudes indicating positions along the *x* and *y* directions, respectively (Fig. 6a). We chose this task because the oscillatory output requires more complicated timing and output dynamics than a reaching task, and is reminiscent of past studies in motor control[48–50].

We manipulated this new task to test the generalizability of our previous results. First, we previously showed that having a larger motor repertoire can facilitate adaptation by creating intermediate trajectories in activity space (Fig. 6a). To test this, we trained the networks on two sets of repertoires with different numbers of movements (Fig. 6b). In one set of repertoires, networks learned one to four movements consisting of cosine waves of different amplitudes, but of sine waves of the same amplitude, creating ellipses of different elongations in the *x* direction. The other set of repertoires was similar, but with sine waves of different amplitudes and cosine waves of the same amplitude. Second, we previously showed that adaptation is not only influenced by the motor repertoire but also how it was initially learned: to facilitate adaptation, the structure between the inputs, neural space, and the changes imposed by the perturbation all have to be congruent. To test the effects of different learning experiences, we encoded the target signal of the sine wave as a continuous input representing its amplitude, and the target signal of the cosine wave as a categorical one-hot encoded input that represented the same amplitude values (Fig. 6a). To test the effects of different perturbations, we asked all networks to adapt to: (1) a 'sine wave perturbation' where they had to produce a movement with a larger sine wave amplitude, (2) a 'cosine wave perturbation' where they had to produce a movement with a

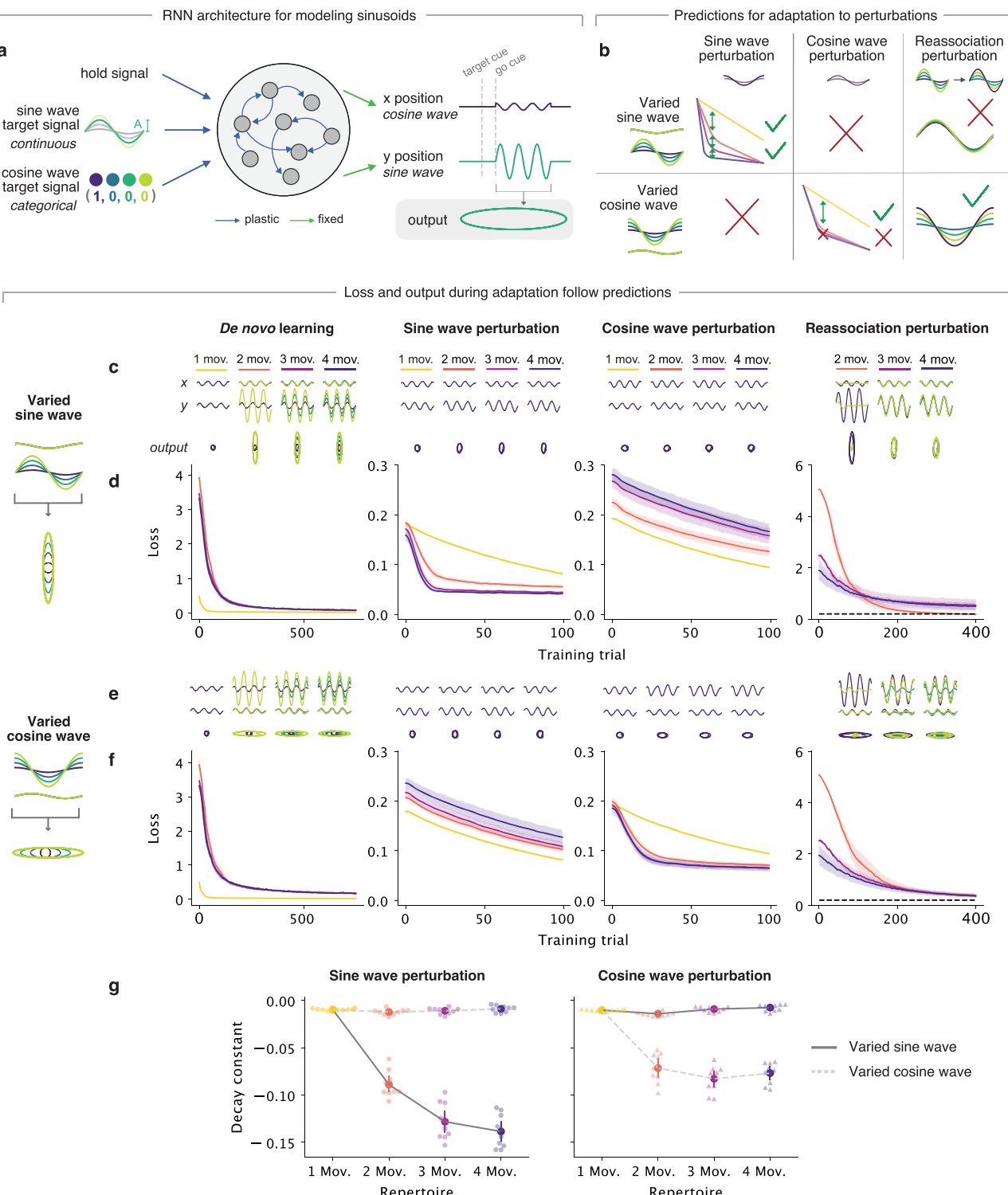

**Fig. 6 | The observed joint influence of motor repertoire and de novo learning experience extends to a more complex sinusoidal task. a** Networks were trained to produce ellipses with up to four different geometries by generating cosine and sine waves representing $x$ and $y$ positions, respectively. Networks were given a hold signal that indicated movement initiation, a continuous sine wave target signal that indicated the amplitude of the sine wave, and a categorical 'one-hot-encoded' cosine wave target signal. **b** Networks were trained on repertoires with different numbers of movements (from one to four) to model de novo learning. Within each repertoire, the movements could consist of either different sine waves and a constant cosine wave ('varied sine wave'), or different cosine waves and a constant sine wave ('varied cosine waves'). Following de novo learning, networks were separately trained to counteract (1) a sine wave perturbation, (2) a cosine wave perturbation, and (3) a reassociation perturbation where they had to produce different movements given learned target

cues. The schematic indicates our predictions of what perturbations would be easier to counteract based on the center-out results. **c** Cosine waves ($x$ position), sine waves ($y$ position), and resultant elliptical output produced by an example network trained to produce sine waves with various amplitudes after de novo learning and each of the perturbations. **d** Loss during de novo learning and adaptation to each of the perturbations. Line and shaded surfaces smoothed mean and 95% confidence interval across networks of different seeds ($n = 10$ random seeds). Dashed line, loss of 0.2 included for easier comparison between loss in different networks. **e, f** Same as panels (**c, d**) but for networks trained on repertoires with varied cosine waves. For the reassociation perturbation, 45% of networks trained on either three- or four-movement repertoires with varied sine waves were able to adapt (MSE < 0.4), compared to 70% for those with varied cosine waves. **g** Decay constants for exponential curves fitted to the loss curves in panels (**d, f**) for the sine and cosine wave perturbations.

larger cosine wave amplitude, or (3) a 'reassociation perturbation' where they had to re-associate target cues to different movements (Fig. 6b).

We could make several predictions based on our previous results (summarized in Fig. 6b). Regarding the networks that learned varied sine waves, we first predicted that networks with larger repertoires would adapt faster to the sine wave perturbation since the continuous amplitude input encoding of the sine wave is congruent to the continuous amplitude change required by the perturbation. Second, these networks would not adapt faster to the cosine wave perturbation since they have not learned multiple cosine waves and thus do not have the intermediate states necessary to facilitate adaptation. Third, these networks would not adapt well to the reassociation perturbation since their continuous inputs are incongruent with the categorical changes required by the perturbation. Regarding the networks that learned varied cosine waves, we first predicted that they would not adapt faster to the sine wave perturbation since they have not learned multiple sine waves. Second, networks with multi-movement repertoires would adapt faster to the cosine wave perturbation than those with single-movement repertoires since they have some structure. However, larger multi-movement repertoire networks would not adapt faster since the categorical input encoding of the cosine wave is incongruent with the continuous amplitude change required by the cosine wave perturbation. Third, these networks would adapt well to the reassociation perturbation since their categorical inputs are congruent to the categorical changes required.

The results support all of our predictions (Fig. 6c–g), showing that our findings on the center-out reaching task generalize to more complicated movement settings. Specifically, different components of a given movement—here, the $x$ and $y$ oscillatory components—can be learned using different input cues, and the way in which these movements are learned can lead to contrasting trends in adaptation to different perturbations. Therefore, our observation that the acquired motor repertoire and the de novo learning process jointly shape the structure of the neural activity to influence subsequent adaptation holds for more complex tasks than center-out reaches.

## Discussion

Adapting to an external perturbation may require the generation of new activity patterns whose availability is likely shaped through long-term motor skill learning. Motor repertoires from different learning experiences thus provide different initial conditions for adaptive activity to evolve. Here, we examined how the learned motor repertoire and how it is acquired jointly shape the underlying structure in neural space and consequently impact motor adaptation using RNNs. We hypothesized that having a larger repertoire of movements could facilitate adaptation since more activity patterns would be readily available, but only if this repertoire was learned in a way that was 'meaningful' for counteracting the perturbation. Indeed, networks with larger repertoires could adapt to perturbations more quickly, but only if the structure of the inputs, the neural space, and the changes imposed by the perturbation were all congruent. These results suggest that adaptation is affected by both the set of existing activity patterns and the organization of these patterns in neural space, which is reflective of the initial de novo learning experience.

Previous work showed that, during a carefully designed BCI adaptation experiment, short-term adaptation could be achieved by re-associating existing neural activity patterns, such that the overall set of neural activity patterns remains the same following adaptation[32]. Other adaptation studies observed that neural activity was shifted following motor adaptation[4,6]. Notably, however, this shift only occurred under force field perturbation, and not under VR perturbations[4,6], suggesting that our modeling results may fall under the former reassociation regime. Indeed, all our multi-movement networks readily had the ability to produce the activity

patterns necessary for adaptation following de novo learning: they were able to adapt to perturbations during adaptation training even with frozen recurrent weights (Supplementary Fig. 9e, f), showing that recurrent weight changes that produce new network dynamics were not necessary. Within this regime, networks with more structure in activity space still adapted more easily, suggesting that the underlying structure determines how easily activity patterns can be deployed during adaptation. This adds an additional layer of complexity to studies investigating adaptation within the existing neural manifold[16,30,37].

Furthermore, we could define a way by which the de novo learning experience shapes subsequent adaptation by establishing a relationship between input structure, neural space structure, and perturbation structure. This relationship lends insight to previous findings. Work on 'structural learning' showed that participants adapted more easily to novel perturbations that have the same structure as perturbations from prior experience[44,51]. They argued based on behavioral data that knowing this structure helps learning by facilitating exploration of a previously acquired low-dimensional 'task-related' space. Here, we showed that the initial de novo learning phase creates structure in neural space by organizing the acquired latent trajectories in a manner reflective of the learning experience, imposing constraints on the trajectory and even the feasibility of future learning (Figs. 4–6). Contextual information is another way by which the learning experience can be shaped[52], and generalization in motor adaptation is highly dependent on the context[53]. In our model, different types of contexts were created via different types of inputs, and these manipulations led to predictable differences in future learning experiences based on the congruency between the inputs and the changes required by a specific perturbation (Figs. 4–6). Thus, our work suggests a potential common neural basis for various behavioral observations spanning structural learning, generalization of motor adaptation, and context inference.

We can make several experimental predictions based on our results. First, while it is experimentally difficult to examine long-term learning on the timescales we are interested in, we may be able to test our predictions in long training sessions that employ repeated movements. Verstynen and Sabes[54] showed that participants had more varied reach angles towards a given target when a series of recently performed reaches had more variance. These experience-dependent changes in variance mirror those we saw in our simulations (Supplementary Fig. 2), where networks that learned more movements had greater variance and less precision in the motor output (Fig. 2e). These similarities suggest that we could perhaps use longer training sessions to study some effects of long-term learning on short-term adaptation. Thus, our modeling results (Fig. 2e, f) predict that there would be a trade-off between motor output precision and robustness to neural noise[47] when more movements are learned through repetition using a paradigm similar to that in the study of Verstynen and Sabes[54].

Second, while there were some differences between multi-movement networks of different sizes, the greatest differences were between single and multi-movement networks (Fig. 2). Multi-movement networks with the same distribution of learned movements had similar ease in adaptation (Fig. 3b, c and Supplementary Fig. 5d, e), since networks were able to produce activity for intermediate movements within the distribution. Thus, we predict that participants would adapt more quickly if they learned a larger distribution of movements, and learning fine-tuned movements within the distribution would provide smaller benefits (provided that these were acquired through the same learning process).

Third, the de novo learning experience, which we modeled by manipulating the structure of the inputs, shaped how the networks adapted under different perturbations by changing the underlying structure in neural space (Figs. 4–6). By comparing networks and

monkeys trained on the center-out reaching task, we saw that the deviation angles in the monkey neural activity were more similar to those for networks with angular inputs (Supplementary Fig. 8e). Classically, the center-out reaching task has been performed by showing participants the position of the reach target[6,55–57]. This is most similar to presenting angular inputs since the cues specify the angular location of the target. Assuming angular inputs in data from monkeys engaged in a VR adaptation task, the speed of within-session adaptation was robustly correlated with the degree of congruency between the structure across the inputs, neural activity space, and the perturbation, giving further experimental support to our modeling result that congruency facilitates learning (Supplementary Fig. 8f). To further experimentally examine the potential impact of the structure of the inputs on learning, categorical inputs could be used instead by cuing targets with different shapes or colors rather than the explicit target location. Our results suggest that learning the center-out task based on these two different types of cues would lead to analogously different patterns of adaptation under different kinds of perturbations, such as a VR or the reassociation perturbation we have studied.

There are several limitations and ways to expand on our work. To assess how different motor repertoires affect adaptation, we trained the networks in two stages: de novo learning of multiple movements followed by adaptation on a single movement. This set-up makes the networks vulnerable to catastrophic forgetting, since the networks may forget the other movements that are not being trained on during adaptation. Training with FORCE learning[58] was especially susceptible to catastrophic forgetting: following adaptation, networks were largely unable to produce output for the other movements (Supplementary Fig. 6). In contrast, training with stochastic gradient descent, a technique known to replicate various experimental results in adaptation[29,59], was largely able to overcome catastrophic forgetting (Supplementary Fig. 6) without overwriting the initial network (Supplementary Fig. 11). Consequently, we decided to use stochastic gradient descent throughout our simulations since it was more behaviorally relevant.

By directly relating the network model activity to motor output, we aimed to model population activity in the motor cortex, which in primates is the main cortical area that projects to the spinal cord to cause movement[60]. While the model has similarities to experimental recordings in the monkey motor cortex (Supplementary Fig. 8), it has not been explicitly fitted to neural data[61–63]. Thus, our model is largely region-agnostic, and it is still unclear where these neural changes due to motor learning and adaptation may occur. Motor learning seems to be associated with activity changes across cortical regions such as the premotor cortex[5,6,29,64], primary motor cortex[29,65,66], parietal cortex[67,68], as well as other structures such as the cerebellum[69–71] and perhaps even basal ganglia[72–75]. How these different regions interact to affect skill learning and adaptation is an ongoing area of study, and future work could use modular and area-specific networks that are constrained by neural data to tease apart their contributions[61–63].

Another potential direction of future research is to extend our current results on the structure of the latent trajectories in neural space to an investigation of the 'dynamical motifs' underpinning these trajectories[8,76]. Since different dynamical motifs arise from learning different tasks[76], different motifs may similarly arise based on different learning processes. Given that the time-varying latent dynamics mediate the planning and execution of movement and exhibit changes during adaptation[6], these dynamical motifs may then predict the result of adaptation to different perturbations.

Future work can also examine more complex motor skills and learning experiences. Here, we have primarily focused on simple movements adapted from the center-out reaching task since it allowed us to compare the motor output and latent dynamics to experimental data—although we have reproduced our key results in a more complex task that simulated tracking ellipses with various geometries (Fig. 6). We can expand on this work to examine more complex and realistic repertoires that include different behaviors like grasping and manipulation, along with models that consider arm kinematics[77,78] and dynamics[79]. Different actions have been shown to occupy different parts of neural space[13,15], so different combinations of behaviors are likely to require different underlying manifolds, which would affect subsequent adaptation to perturbations on any given behavior. In addition, we can examine how learning is affected by different training processes. Here, we have specified that our networks undergo de novo learning since they are trained to perform reaches for the first time, but our results may not be specific to this situation. Rather, our results may be more indicative of how prior training affects further learning in general. To probe this further, future work can examine how networks pretrained on many tasks learn to perform a new skill, compared to networks that are trained from scratch on the same new skill or networks that are trained on different similar skills in sequence.

In conclusion, we have shown that de novo motor learning shapes subsequent adaptation based on the structure it creates in neural space. Most dramatically, two sets of networks that have different neural structures can exhibit opposite trends when adapting to the same perturbation, even if they know the same set of movements equally well. The neural structure is defined by the specific de novo learning process, which can be manipulated by providing cues that either have or lack a meaningful relationship among them. Furthermore, knowing a larger repertoire of movements often facilitates adaptation, but only when the changes required by the perturbation are congruent with the structure of the underlying latent dynamics. While we have examined the formation of structure in neural space in the context of motor learning, similar structural constraints may arise in other systems, shaping not only motor but also cognitive processes.

## Methods

### Reaching task

We trained recurrent neural networks to perform a standard center-out reach task, which is commonly used in experimental settings to examine motor control. We modified existing experimental data for reaches using the task design of ref. 6. In the experiment, a monkey controlled a cursor on a computer screen using a 2D manipulandum. The cursor started in the center of a circle with a radius of 8 cm, and monkeys had to reach one of eight possible targets spaced around the circle. The monkeys were shown which target to move to at a target cue, but they had to delay movement until a later go cue. To understand how existing skill sets affect motor adaptation, we created four motor repertoires of different sizes, ranging from one to four reach movements to one to four targets, respectively. We adapted the experimental reaches of one monkey, Monkey M, from ref. 6 to create the target reaches for each movement by rotating the experimental reaches to different targets that we defined. In our case, the targets were equally spaced around an arc of the circle. Unless specified otherwise, the arc spanned from −10° to −50°. Networks were trained on one of these four repertoires. By using different numbers of movements in each skill set, we were able to examine how a network that knows more movements may adapt differently than a network that knows less. Each trial lasted 4.0 s: the target and go cues were randomly selected for each trial, with the target cues being presented between 1.0–2.5 s after the beginning of the trial, and the go cues between 2.5–3.0 s after the beginning of the trial.

To examine network performance without trial-to-trial movement variability, which was inherent in experimental reaches, we also created repertoires with synthetic reaches. These synthetic reaches had fixed target and go cues based on the mean target and go cues in the experimental data, and the target position profiles of each reach

was the same across trials. Each synthetic reach lasted 1.0 s and the position profile was defined by a sigmoid for each time $t$ for the length of the reach $l = 8$ cm:

$$\frac{l}{1 + e^{-12t+6}} \qquad (1)$$

To assess motor adaptation, we examined how networks adapted to a visuomotor rotation (VR), a common perturbation used in experimental settings. To simulate VR, we rotated the output position of the network counterclockwise by $\theta_r$ during adaptation trials. We used rotations with $\theta_r = 10°$, 30°, or 60°. We also examined how networks adapted to visuomotor re-associations where the target cues and targets are rearranged, such that the network must reach a different known target given a known target cue.

## Sinusoidal task

To extend our results to a more complex task, we trained recurrent neural networks to produce cosine and sine waves representing $x$ and $y$ positions that trace out ellipses. The task followed a similar trial structure to that of the delayed reaching task: each trial lasted 4.0 s, and the target cues were fixed at 800 ms while the go cues were fixed at 1300 ms. We created synthetic movements that lasted 2.0 s after the go cue. Each movement consisted of a triple sine wave and a triple cosine wave, which traced out three cycles of an ellipse. To understand how existing repertoires affect motor adaptation, we created two sets of repertoires of different sizes, ranging from one to four movements tracing out one to four ellipses, respectively. For the first set of repertoires, we varied the amplitude of the sine waves such that different movements in each repertoire had sine wave amplitudes equally spaced within a range of 1 to 7 cm, and a constant cosine wave with an amplitude of 1 cm. The single-movement repertoire had a sine wave amplitude of 1 cm. For the second set of repertoires, we varied the amplitude of the cosine waves instead, while keeping the sine wave amplitude constant.

To assess motor adaptation, we examined how networks adapted separately to three different perturbations. First, we asked networks to adapt to a 'sine wave perturbation' on the one movement shared by all repertoires (the one with sine and cosine wave amplitudes of 1 cm) by producing a sine wave with a larger amplitude of 2 cm. Second, we asked networks to adapt to a 'cosine wave perturbation' on the one shared movement by producing a cosine wave with a larger amplitude of 2 cm. Third, we asked networks to adapt to a reassociation perturbation where the target cues and target movements were rearranged, such that the network must produce a different movement given a known target movement cue.

## Neural network model

**Network architecture.** The model dynamics were given by:

$$\tau \dot{x}_i(t) = -x_i(t) + \sum_{j=1}^{N} J_{ij} r_j(t) + \sum_{k=1}^{I} B_{ik} s_k(t) + \eta_i(t) \qquad (2)$$

where $x_i$ is the hidden state of the $i$th unit and $r_i$ is the corresponding firing rate following $tanh$ activation of $x_i$. The networks had $N = 300$ units and $I$ inputs. The time constant $\tau$ was set to 0.05 s, the integration time step $dt$ to 0.01 s, and the noise $\eta$ was randomly sampled from the Gaussian distribution $\mathcal{N}(0, 0.2)$ for each time step. The initial states $\mathbf{x}_{t=0}$ were sampled from the uniform distribution $\mathcal{U}(-0.1, 0.1)$. The networks were fully recurrently connected, with the recurrent weights $\mathbf{J}$ initially sampled from the Gaussian distribution $\mathcal{N}(0, \frac{g}{\sqrt{N}})$, where $g = 1.2$. The time-dependent stimulus inputs $\mathbf{s}$ (specified below) were fed into the network, with input weights $\mathbf{B}$ initially sampled from the uniform distribution $\mathcal{U}(-1.0, 1.0)$.

For the networks trained on the reaching task, two types of sustained inputs $\mathbf{s}$ were used. For the angular inputs, $\mathbf{s}$ was three-

dimensional and consisted of a one-dimensional hold signal and a two-dimensional target signal ($2\,cos\theta^{\text{target}}$, $2\,sin\theta^{\text{target}}$) that specified the reaching direction $\theta^{\text{target}}$ of the target. For the categorical inputs, $\mathbf{s}$ was five-dimensional and consisted of a one-dimensional hold signal and a four-dimensional one-hot encoded target signal with the same magnitude (e.g., (0, 0, 2, 0)) that did not provide information about the target's angular direction. For the networks trained on the sinusoidal task, one type of sustained input $\mathbf{s}$ was used that included both continuous and categorical encoding of target movements. For this task, $\mathbf{s}$ was six-dimensional and consisted of a one-dimensional hold signal, a one-dimensional sine wave target signal ($2A/A_{\text{max}}$, where the maximum amplitude $A_{\text{max}}$ was 7) that specified the amplitude $A$ of the sine wave, and a four-dimensional one-hot encoded cosine wave target signal with the same magnitude (e.g., (0,0,2,0)) that specified the cosine wave target without providing information about its amplitude. For all types of inputs, the hold signal started at 2 and goes to 0 at the go cue, and the target signals remained at 0 until the task cue.

The networks were trained to produce 2D outputs $\mathbf{p}$ corresponding to $x$ and $y$ positions of motor trajectories, and they were read out via the linear mapping:

$$p_i(t) = \sum_{k=1}^{N} W_{ik} r_k(t) \qquad (3)$$

where the output weights $\mathbf{W}$ were sampled from the uniform distribution $\mathcal{U}(-1.0, 1.0)$. During VR adaptation trials, $\mathbf{p}$ was rotated counterclockwise according to a perturbation angle $\theta_r$.

**Training the model.** For the reaching task, networks were optimized to generate positions of reach trajectories modified from ref. 6. The training and testing datasets were created by pooling successful trials during baseline epochs across all experimental sessions for Monkey M (2208 trials: 90% training, 10% test). The experimental data was modified for each repertoire (see Reaching task), and equal numbers of trials for each reach direction were included for each repertoire. For the sinusoidal task, networks were optimized to generate synthetic sine and cosine waves representing elliptical movements without trial-to-trial variability for each movement.

Unless otherwise specified, networks had to learn their input weights $\mathbf{B}$ and recurrent weights $\mathbf{J}$ while the output weights $\mathbf{W}$ remained fixed. To model motor skill learning, we initially trained the networks on repertoires with one to four movements, using the Adam optimizer with an initial learning rate $l = 10^{-4}$, first moment estimates decay rate $\beta_1 = 0.9$, second moment estimates decay rate $\beta_2 = 0.999$, and epsilon $\epsilon = 1e-8$. Then, to model motor adaptation, we trained the pretrained networks to counteract a perturbation (VR or reassociation for the reaching task; cosine wave, sine wave, or reassociation for the sinuoidal task), using stochastic gradient descent with a fixed learning rate $l = 5^{-3}$, unless otherwise specified. We used a faster learning rate during adaptation to model faster short-term learning compared to long-term skill learning. To assess how existing repertoires may shape adaptation on a given movement, networks were only trained to counteract the VR, sine wave, or cosine wave perturbation on the one target that all repertoires shared (i.e., the $-10°$ reach, or the ellipse consisting of cosine and sine waves with an amplitude of 1), such that performance could be compared across networks with different repertoires. Initial training was implemented with 750 training trials and a batch size $B = 64$. For the reaching task, adaptation training was implemented with either 100 training trials for VR perturbations or 300 training trials for reassociation perturbations, and a batch size $B = 64$. For the sinusoidal task, adaptation training was implemented with either 100 training trials for the cosine and sine wave perturbations or 400 training trials for reassociation perturbations, and a batch size $B = 64$. All training configurations were performed on 10 different

networks initialized from different random seeds. To examine adaptation under an alternate learning algorithm, we also trained the networks to counter the VR perturbation during the reaching task using FORCE learning[58] with a learning rate of 100.

The loss $L$ was the mean-squared error between the two-dimensional output and target positions over each time step $t$, with the total number of time steps $T = 400$. The first 50 time steps were not included to allow network dynamics to relax:

$$L = \frac{1}{2B(T-50)} \sum_{b=1}^{B} \sum_{t=50}^{T} \sum_{d=1,2} \left( p_d^{\text{target}}(b,t) - p_d^{\text{output}}(b,t) \right)^2. \quad (4)$$

To produce dynamics that aligned more closely to experimentally measured neural dynamics[38,39], we added L2 regularization terms for the activity rates and network weights in the overall loss function $L_R$ used for optimization:

$$L_R = L + R_W + R_R \quad (5)$$

where

$$R_R = \frac{\beta}{BTN} \sum_{b=1}^{B} \sum_{t=0}^{T} \sum_{n=1}^{N} r_n(b,t)^2 \quad (6)$$

and

$$R_W = \alpha(||\mathbf{J}|| + ||\mathbf{B}|| + ||\mathbf{W}||) \quad (7)$$

where $\beta = 0.5$ and $\alpha = 0.001$. Note that the loss recorded in the main text was the loss $L$ before regularization, and it was smoothed with a backward moving average of five trials. We clipped the gradient norm at 0.2 before applying the optimization step. While we used the specific parameters described above for training our neural network models, we also trained the models with varied parameters and the trends in our main results remain generally robust to these changes (Supplementary Fig. 12).

## Data analysis
Analyses on the neural activity were examined for both the preparation and execution epochs of the movement, taken as 500 ms before and 1000 ms after the go cue, respectively.

To assess how latent dynamics change during motor learning and adaptation, we examined the neural space of the networks. In the neural space of a population of $n$ units, each axis corresponds to the firing rate of a unit, and each point denotes the state of the neural population at a certain time step. To obtain smooth firing rates through time, we applied a Gaussian kernel (std = 50 ms) to the activity rates from the networks. We identified a lower $m$-dimensional neural manifold in the activity space by applying PCA to the smoothed firing rates of the units. PCA finds $n$ orthogonal basis vectors (principal components or PCs) that maximally capture the variance in the population activity, sorted by their corresponding eigenvalues. We kept the $k$-leading PCs, or neural modes, that captured the majority of the variance: $k = 10$ captured more than 80% of the variance in our network activity. We projected the original smoothed firing rates onto the neural modes to get the latent dynamics of the networks.

To measure the variability in activity and motor output, we aligned trials by the go cue and measured the variance across trials for the same reach at corresponding time points for each feature (i.e., for each unit, latent dimension, or output dimension). To compare the variance in latent dynamics across different neural spaces, we first normalized the latent dynamics by the median distances between trial-averaged time points within the neural space before calculating the variance. Variance in unit activity and latent dynamics was calculated

during both preparation and movement, while variance in output position was calculated during movement. Reach angles were calculated based on the mean angle for the entire reach during movement, and the variance was calculated across trials. To determine how patterns in variance in the motor output differed from those in the activity, we examined the variance in unit activity in the output-potent and output-null subspaces[46]. Unit activity was directly related to the motor output through the read-out weights $\mathbf{W}$ of the networks. The output-potent dimensions were then the row space of $\mathbf{W}$, while the output-null dimensions were the null space of $\mathbf{W}$. We projected the unit activity onto these dimensions to obtain the activity in the respective subspaces. For each measure, we calculated either the median variance across all features and time points (Fig. 2) or the median total variance summed over all features, across all time points (Supplementary Fig. 3). Notably, the trends remained the same for the different variance calculations.

To measure the precision of the latent and motor trajectories, we calculated the 'tangling' in both the latent space ($Q_x$) and output space ($Q_p$), adapted from the measure specified in ref. 80:

$$Q_x(t) = \max_{t'} \frac{||\dot{\mathbf{x}}_t - \dot{\mathbf{x}}_{t'}||^2}{||\mathbf{x}_t - \mathbf{x}_{t'}||^2 + \epsilon_x} \quad (8)$$

$$Q_p(t) = \max_{t'} \frac{||\dot{\mathbf{p}}_t - \dot{\mathbf{p}}_{t'}||^2}{||\mathbf{p}_t - \mathbf{p}_{t'}||^2 + \epsilon_p} \quad (9)$$

where $\mathbf{x}_t$ is the latent activity at time $t$, $\dot{\mathbf{x}}_t$ is its temporal derivative, $\mathbf{p}_t$ is the position at time $t$, $\dot{\mathbf{p}}_t$ is its temporal derivative, $|| \cdot ||$ is the Euclidean norm, and $\epsilon_x$ and $\epsilon_p$ are small values equal to 0.1 times the average squared magnitude of $\mathbf{x}_t$ and $\mathbf{p}_t$, respectively, to prevent division by zero. Tangling measures how dissimilar future states can arise from similar current states, so it can be used to examine the precision of different trajectories. Note that tangling is reported as the 90th-percentile across all time steps in the main text.

To assess how activity changed during motor adaptation, we visualized these changes by projecting both the unit activity before and after adaptation onto the neural modes of the manifold calculated before adaptation (Fig. 4b, e). We further quantified the manifold overlap before and after adaptation training as specified in ref. 31, adapted from ref. 81. To find the manifold overlap between the manifolds of two network activities $\mathbf{A}_1$ and $\mathbf{A}_2$, we first calculated the covariance matrix $\mathbf{C}_1$ of $\mathbf{A}_1$ and projected it onto its neural manifold identified through PCA. We then calculated the covariance matrix $\mathbf{C}_2$ of $\mathbf{A}_2$ and projected it onto the neural manifold of $\mathbf{A}_1$. To quantify the variance explained by these projections, we divided the trace of these projections by the trace of the corresponding covariance matrices:

$$\beta_1 = \frac{Tr(\mathbf{V_1 C_1 V_1^T})}{Tr(\mathbf{C_1})} \quad \beta_2 = \frac{Tr(\mathbf{V_1 C_2 V_1^T})}{Tr(\mathbf{C_2})} \quad (10)$$

where $\mathbf{V}_1$ are the first ten principal components resulting from PCA on $\mathbf{A}_1$. Here, $\beta_1$ is the variance in $\mathbf{A}_1$ that can be explained by the neural manifold for $\mathbf{A}_1$ while $\beta_2$ is the variance in $\mathbf{A}_2$ that can be explained by the neural manifold for $\mathbf{A}_1$. We then calculate the manifold overlap as the ratio $\beta_2/\beta_1$.

Changes in the neural manifold were driven by changes in synaptic connectivity, so we also measured the relative weight change $\mathbf{dJ}$ for the recurrent weights before ($\mathbf{J}_{\text{before}}$) and after ($\mathbf{J}_{\text{after}}$) adaptation training:

$$\text{relative } \mathbf{dJ} = \left| \frac{\mathbf{J}_{\text{after}} - \mathbf{J}_{\text{before}}}{\mathbf{J}_{\text{before}}} \right|. \quad (11)$$

where $|\cdot|$ is the absolute value and the division is element-wise. To examine the direction of these changes in the manifold relative to the initial shape of the manifold, we defined a metric called the 'deviation angle'. First, we defined 'adjacent movement vectors' $\mathbf{v}_{adj}$ between corresponding time points of the trial-averaged latent trajectories of the first movement $\mathbf{x}_{m_1}$ and its adjacent movement $\mathbf{x}_{m_2}$ (the second movement) before adaptation:

$$\mathbf{v}_{adj}(t) = \mathbf{x}_{m_2}(t) - \mathbf{x}_{m_1}(t) \tag{12}$$

These vectors quantified the general shift from one movement to the next in neural space and approximated the shape of the manifold between the adjacent movements. Then, we performed a similar computation for the first movement before ($\mathbf{x}_{m_1}$) and after ($\tilde{\mathbf{x}}_{m_1}$) adaptation to get the 'adaptation vector' $\mathbf{v}_{adp}$ that quantified the general shift during adaptation:

$$\mathbf{v}_{adp}(t) = \tilde{\mathbf{x}}_{m_1}(t) - \mathbf{x}_{m_1}(t). \tag{13}$$

We defined the 'deviation angle' as the angle between these two vectors, which measures how changes in adaptation deviated from the path afforded by the existing structure in neural space before adaptation.

To examine congruency between the structure of the inputs, neural space, and the changes required by the perturbation, we defined congruence as the representational similarity between any two structures. For the input structure, we constructed a representational dissimilarity matrix (RDM)[82] by calculating the cosine dissimilarity between pairs of target signal input vectors associated with each movement. For the neural space structure, we constructed an RDM by calculating the distances between the latent trajectories associated with each movement. Specifically, we quantified the median distance $D$ between the trial-averaged latent dynamics $\mathbf{x}_1$ and $\mathbf{x}_2$ for each pair of movements over all corresponding time points $t$:

$$D = \mathrm{med}(||\mathbf{x}_1(t) - \mathbf{x}_2(t)||). \tag{14}$$

To compare these distances between movements across different neural spaces, we normalized by the median distances between time points within each movement, pooled across all movements $m$:

$$\hat{D} = \frac{D}{\mathrm{med}(||\mathbf{x}_m(t) - \mathbf{x}_m(t-1)||)}. \tag{15}$$

We could then measure the congruence between the input and neural space structures by calculating Pearson's correlation coefficient between the upper triangular regions of their respective RDMs. The structure of the changes required by the perturbation was harder to define mathematically since the nature of the perturbation can vary widely. Instead, we described the congruence qualitatively: an angular perturbation like a VR perturbation is congruent to angular inputs, and a categorical perturbation like a reassociation perturbation is congruent to categorical inputs.

### Experimental comparison

To verify that our networks produced realistic latent dynamics, we trained an additional set of networks on the original 8-target center-out reach task from ref. 6 (see Reaching task). The target trajectories for training and testing were based on reach trajectories during successful trials pooled from all sessions for Monkey C (see Training the model). Following training, we compared the simulated activity to experimental activity recorded from motor cortex during one baseline session for Monkey C[6]. We pre-processed the experimental recordings by removing units with trial-averaged firing rates less than 5 Hz and applying a Gaussian kernel (std = 50 ms) to the binned square-root transformed firings of each unit (bin size = 30 ms). We then subtracted the cross-condition mean. To compare the latent dynamics (see Data Analysis), we used Canonical Correlation Analysis (CCA), which finds new directions in the neural manifold that maximize the pairwise correlations between two datasets when they are projected on these directions[83]. Canonical correlation values range from 0 to 1, with 1 being a complete correlation.

To verify the 'deviation angle' metric, we trained the networks to counteract a 30° VR perturbation on all eight targets and compared the deviation angles to those found in monkeys performing the same adaptation (3 sessions for Monkey C, 3 sessions for Monkey M, 10 sessions for Monkey M2; Monkey M and M2 are the same monkey, but they are treated separately since the recordings are from different hemispheres in motor cortex and were collected while the monkey performed the task with the contralateral arm over two separate sets of experiments). Deviation angles were found for the latent dynamics corresponding to reaches for all 8 targets. We also calculated deviation angles for shuffled targets and time points as a control for the networks.

To examine congruency between the structure of the inputs and neural space, we assumed that monkeys were given angular inputs during the center-out reach task since they were shown the location of the target centered around a circle. We calculated the congruency between the structure of angular inputs and the structure of their neural spaces. To compare congruency with the extent of adaptation, we calculated error curves for each session based on the angular error of monkey reaches in the first 150 ms and calculated decay constants for exponential curves fitted to these error curves, as we did for the loss curves during training for the networks. Note that sessions where the monkeys did not adapt sufficiently (i.e., sessions where exponential curves failed to fit the error curves) were excluded from all analyses.

### Reporting summary

Further information on research design is available in the Nature Portfolio Reporting Summary linked to this article.

## Data availability

We did not acquire new monkey datasets for experimental comparison, and instead relied on existing datasets from ref. 6. Most of the monkey datasets used have been previously analyzed[6,42,84,85] and made publicly available on Dryad (https://doi.org/10.5061/dryad.xd2547dkt). The remaining datasets will be made available on request.

## Code availability

All analyses were implemented using custom python code (Python 3.6) and open-source software. All the figures are reproducible by running Jupyter notebooks. Code to reproduce all the results is openly available at https://github.com/JoannaChang/de_novo_learning_structure.

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

## Acknowledgements

J.C.C. received funding from the Wellcome Trust (grant 108908/Z/15/Z). M.G.P. received funding from the Fonds de recherche du Québec Santé (grant chercheurs-boursiers en intelligence artificielle J1). L.E.M. received funding from the NIH National Institute of Neurological Disorders and Stroke (NS053603 and NS074044). J.A.G. received funding from the EPSRC (EP/T020970/1) and the European Research Council (ERC-2020-StG-949660). C.C received funding from the BBSRC (BB/N013956/1 and BB/N019008/1), the EPSRC (EP/R035806/1), the Wellcome Trust (200790/Z/16/Z), and Simons Foundation (564408). The funders had no role in study design, data collection and analysis, decision to publish, or preparation of the manuscript.

## Author contributions

J.C.C., J.A.G., and C.C. devised the project. M.G.P. and L.E.M. provided the monkey datasets. J.C.C. ran simulations, analyzed data, and generated figures. J.C.C., C.C., and J.A.G. interpreted the data. J.C.C., C.C., and J.A.G. wrote the manuscript. All authors discussed and edited the manuscript. J.A.G. and C.C. jointly supervised the work.

## Competing interests

J.A.G. receives funding from Meta Platform Technologies, LLC. The remaining authors declare no competing interests.
