## [Peer Review File · Nature Communications]

De novo motor learning creates structure in neural activity space that shapes adaptationREVIEWER COMMENTS

Reviewer #1 (Remarks to the Author):

The authors use a recurrent neural network model to explore how prior training/learning (“de novo motor learning”) may influence subsequent learning dynamics (“adaptation”). They do so by training naive RNN models on different initial motor tasks and then seeing how that initial training influences the model’s ability to adapt to a perturbation of this task. This setup allows the authors to systematically control for training experience (e.g., using the exact same initial model and only changing the training history), which cannot be readily done in experiments. Through analyses and model variations, the authors find that the initial task training influences adaptation and candidate sources of this influence: changes in the number of neural activity patterns the model generates and the structure of these activity patterns. One of the more compelling and novel findings in the study is a simple demonstration that how neural activity patterns are structured in relation to input cues influences how the model can (or cannot) adapt to future perturbations. The study brings computational RNN methods to bear on questions about how population neural activity shapes motor learning, which has primarily been explored experimentally. The study therefore provides new perspectives on existing studies in motor and brain-computer interface learning and will be of interest to those communities.

Overall, I find the manuscript to be clearly written, the work thorough and sufficient to back up the primary claims being made. I do have a few comments and suggestions for areas to improve the paper and some technical questions/points of confusion that should be addressed before the paper is suitable for publication.

1. Some phrasing in the introduction and abstract should be refined to reflect the current literature more accurately with regard to the relationship between network connectivity and neural activity patterns. Much of the manuscript’s motivation focuses on a line of research investigating “neural manifolds”. While this literature often emphasizes the interpretation that neural manifolds reflect the underlying anatomical connections within a network of neurons, these studies almost exclusively use correlations, which is a measure of functional connectivity. The relationship between structural and functional connectivity metrics is a significant outstanding challenge in neuroscience (e.g., Das & Fiete, *Nature Neuroscience* 2020). The authors must improve the care in how they discuss this work and avoid conflating functional and structural connectivity. For example, the authors cite Sadtler et al. *Nature* 2014 and Oby et al. *PNAS* 2019 as providing evidence that “this neural manifold is likely shaped, at least partly, by the underlying connectivity of the network”. Those studies show that neural correlation structures influence learning behavior – they do not prove a causal relationship between the neural manifold and the underlying structural network. Similarly, the authors later cite the Oby paper to defend the statement that generating new activity patterns necessitates changing synaptic connectivity – the paper does not directly prove this point. Indeed, past RNN modeling work highlights that BCI learning dynamics may not be purely reflect the degree of synaptic weight changes required (Feulner & Clopath, *PLoS Comp Biol* 2021), highlighting the complexity of neural activity pattern – structural connectivity relationships.

2. I am rather unclear on how the variance calculations for figure 2 were performed. My largest source of confusion is that your potent and null space variances do not appear to add up to your total variance according to all the panels shown in 2c. I could not piece together from the methods how this would be the case. Please more fully explain the calculation methods. I would also recommend more precision in the language used in this section of the manuscript since “variance” can be calculated so many ways and has many different potential interpretations (it wasn’t clear to me until going all the way to the methods that this was trial-to-trial variance, for instance).

3. Given that your different models don’t generate the same overall behavior (e.g. figure 2E and figure S2), I think it would be beneficial to present an analysis showing that this is not purely an artifact of things like training duration. The loss curves suggest all the models have converged similarly, but spelling out that you see this consistent difference independent of total training would make the point more convincing.

4. I am not sure that the paper benefits from using the term “de novo learning”, since ultimately the question being tested is whether past learning/experience influences future learning. I

understand the utility of using existing terms in the motor literature and they are broadly used in accordance with the literature (acquiring a new skill being thought of as different from adapting an existing skill). But is the fact that the RNN is learning reaches for the first time the real point being made? The manuscript does not perform tests to somehow distinguish de novo learning from general prior training. I would suggest that it might be more appropriate to restrict the claims to initial or prior training/learning influencing subsequent adaptation. The potential relationship to concepts of de novo learning seems like a useful and interesting discussion point.

Minor/specific points:

1. Structurally, I find it a bit odd that the manuscript includes a conceptual summary figure (Fig. 5) before they have finished making all the results-based claims. Figure 6 is important for making their claim about "congruent structure", so having that claim in figure 5 seems like the cart before the horse.
2. In the text, being sure to clarify that "congruent structure" is with respect to the perturbation would be helpful – this isn't always clear.
3. What is the dashed line figure panels 6B and 6E?
4. It's not entirely clear to me how panel 6D supports the statement that "neighboring trajectories interfered with one another during adaptation such that the adapted motor output become overlapped". Overlap is not very visually obvious in D (the neural space).
5. While the conceptual labels above groups of panels is clever, they occasionally cause some confusion/challenges. Figure 4K is conceptually related to panels I-L but is given its own distinct label, presumably in part because of its physical location.
6. I do not understand why Figure 2E needs an inset – can't the experimental data point be put on the same axes as the main figure? You could use a discontinuous y-axis if the scale of the data would obscure details of the model data.

Reviewer #2 (Remarks to the Author):

Summary

Animals can adapt their movements to changes in their bodies or environments, but it is an open question why some adaptations are easier than others. This is a profound question with wide implications. Here, the authors take up a longstanding hypothesis – that the history of learned movements (one's 'repertoire') constrains what is learnable – and address it by analyzing a neural network with a known learning history. Applying many of the same analyses applied to animals to this neural network, the authors demonstrate a range of empirical phenomena, some of which match experimental findings (such that adaptations are easier when they align with the dimensions of largest variance in neural space) and some that are new predictions. A central claim is that having wider repertoires facilitates adaptation but only when the required adaptation is 'congruent' with the training history (i.e. sharing an abstract task structure).

Overall, this work tackles an important problem with conceptual originality. It is a topic I find deeply interesting. However, I was left feeling like I did not yet gain a deeper theoretic or mechanistic understanding of how and why the repertoire affects adaptation ability. Without a mechanistic understanding, it is left unproved how general these results are. This could likely be addressed with further analyses and especially by linking to literature in machine learning, as I describe below.

Strengths

I want to commend the authors for an excellently written manuscript and their choice of question. The introduction was a pleasure to read. The citations were all chosen carefully, and they are used well to position this paper within recent literature. Many very interesting subfields are brought together in this work.

The preparation of figures was done thoroughly, as well. Although other settings should also be analyzed, the figures and supplemental figures represented a thorough documentation of the chosen network.

Weaknesses

Where this study does not yet feel complete is in 1) helping to understand **why** these phenomena are observed, and 2) how general are these phenomena to neural networks in other settings. (These are of course related, as an understanding of why this happens would imply its generality).

Neural networks are a powerful model system for this set of questions not only because they recapitulate biological phenomena, but also because they (potentially) allow a deeper understanding as to the underlying mechanisms. This manuscript largely stops at empirical characterization. But why do neural networks find it easier to adapt to movements that interpolate their repertoire? Why does the output variance increase for networks with more diverse training sets? Why does noise robustness increase for such networks, as well? Why is congruence important? This paper provides an interesting set of hypotheses, but it feels as if these need to be verified in future work.

There is a large field of learning theory in neural networks that could help explain these results, but this field is not referenced. The authors are missing a powerful opportunity to leverage such deep learning theory. I would be excited to a review a revision which makes such an attempt.

Relatedly, the current analyses focus rather narrowly on a single setting of just 4 reaching movements, but make rather general conclusions about adaptation (and transfer learning) in neural networks in the abstract. If this is a general truth, it would be of wide interest in the machine learning community as well. However, is not yet clear how far the current conclusions will generalize to other networks or settings, and especially to more complicated movement regimes and networks. Without proven generality it is also hard to extrapolate these results to biological networks.

The generality of these many findings could be proved by empirically evaluating a range of other networks in more complicated movement (or non-movement) settings. (I would especially consider benchmarks in the 'imitation learning' field, which matches this current setting.) Alternatively, generality could be provided by a theoretical understanding of why the current network behaves as it does. (I've provided some suggestions below).

To summarize, this manuscript implies bold claims about transfer learning in neural networks, but does not utilize the extensive literature on this question in machine learning. This has the effect of leaving uncertain why these phenomena are observed or in what other settings they might be expected to occur.

Questions

ll 203-215: That distances in neural space are proportional to distances in motor output would seem to be guaranteed by the use of a linear readout. Differences in proportionality in different settings would then reflect differences in the projection of activity upon the null/potent space of the readout. Would the authors predict that adaptation ability is a consequence of the fraction of preparation/execution activity that projects into the potent space? If preparatory activity is by definition in the null space, this would seem to imply that adaption ability follows having a small variance in preparatory activity? Just trying to follow the logic.

ll 227-238: I'm not yet convinced that this categorical encoding task **proves** adaptation is a consequence of the congruence between neural space and motor output. The categorical encoding task could negatively affect angular adaption for other reasons than the congruence. This seems to be a correlation taken as a causal mechanism. (But why stop there? In neural networks we can know the mechanism and analytically describe learning dynamics.)

ll 275-305: Could the authors provide a more precise definition of 'congruence' between the type of perturbation and the neural structure? Is it possible to define this mathematically? This would help the reader to understand when else to expect adaptation to be easy/hard.

l 495: Were the networks trained with backpropagation through time, or to minimize the objective

at each timepoint (ignoring history)? I'm not getting at anything here, just curious.

I 511: Again just curious, but why were firing rates smoothed with a Gaussian kernel? I don't think it would affect results either way... but I'd bet that running PCA on this timeseries and keeping only lower PCs will result in a lowpass effect anyways.

Suggestions

Some papers the authors may wish to cite are,

Krakauer, J. W., Mazzoni, P., Ghazizadeh, A., Ravindran, R., & Shadmehr, R. (2006). Generalization of motor learning depends on the history of prior action. *PLoS biology*, 4(10), e316.

Sing, Gary C., et al. "Primitives for motor adaptation reflect correlated neural tuning to position and velocity." *Neuron* 64.4 (2009): 575-589.

In Sing et al. (2009), the authors model motor output as a linear combination of the motor repertoire (the 'motor primitives'). (It is interesting to note that this framework resembles some findings in the current work, such as Fig S4, that perturbations can be adapted to if the movement is within the range spanned by learned examples.)

If I may, I'd like to suggest one manner in which the authors might show mathematically that adaptation ability in neural networks relates to the structure of neural activity, and in turn to the history of trained movements. Take or it leave it; there are many ways in which the generality of this paper's claims beyond the single 4-movement setting could be proved.

In modern machine learning parlance, the model of Sing et al. would be called a (linear) kernel machine. What is interesting and powerful about this connection from perspective of adaption is that the learning dynamics of kernel machines can be described exactly. Thus, one could fairly straightforwardly prove what sorts of adaptations are easy/hard to learn. Essentially, the easy perturbations to learn are ones that map onto the largest PCs of the last neural population before the linear combination (the 'kernel'). If this direction interests you, I would recommend the following publications:

Bordelon, B., & Pehlevan, C. (2022). Population codes enable learning from few examples by shaping inductive bias. *Elife*, 11, e78606.

Flesch, T., Saxe, A., & Summerfield, C. (2023). Continual task learning in natural and artificial agents. *Trends in Neurosciences*, 46(3), 199-210.

Additionally, I recommend the authors consult the following publication, which describes analytically how the Sadler et al. result (that BCI tasks are easier when they are aligned with the structure of the learning task) can arise from gradient descent learning.

Humphreys, P. C., Daie, K., Svoboda, K., Botvinick, M., & Lillicrap, T. P. (2022). BCI learning phenomena can be explained by gradient-based optimization. *bioRxiv*, 2022-12.

Reviewer #3 (Remarks to the Author):

This paper uses a recurrent network modeling approach to study how the breadth and type of prior motor learning influences how quickly and accurately the network is able to adapt to perturbations in the motor output. The authors find that networks show faster adaptation when the network has learned a greater breadth of movement angles and when the activity is structured in the network in such a way that it is matched with the type of perturbation. Overall, this is an interesting paper on a timely topic. However, the conditions studied within the network is limited in scope and its relevance to how learning takes place in the brain is currently unclear (see details below).

Major comments:

1) The authors consider only a center-out movement task, and only 4 targets that only span 40 degrees. And the perturbation is to only a single target. It is unclear whether such limited scope allows the broader statements the authors make about how learning influences adaptation in this network. Making a link to the brain is an even bigger stretch. The authors should demonstrate a

wider range of learned tasks by the network (beyond center-out movements), and test how the network adapts subsequently.

2) The authors should examine how their modeling assumptions in L452-458 influence their results. In particular, how did the authors choose the variances (or ranges) of each of the probability distributions from which they drew their noise, initial states, and model parameters? How does varying those assumptions alter the scientific conclusions of the study?

3) As presented, this paper elucidates the learning properties of a recurrent network model. The authors would like to imply that this might be how the brain learns, but at present the links are weak. Real data is contained in Fig 2E and S6, but those figures are more to demonstrate that the order of magnitude of different metrics match between model and data, rather than to show evidence that their main modeling results might hold in the brain. Can the authors make a stronger case using existing experimental data that this is how the brain learns? For example, for the first main result comparing networks which have learned different numbers of movement directions, can the authors dig up data from the Miller lab in which different animals have learned different sets of tasks over the course of their lifetimes, and then compare across animals?

4) Fig 2F: this results seems unexpected. If multi-movement networks have more activity in output-potent subspace relative to output-null subspace compared to a single-movement network, one would expect more noise to "leak" into the movements for multi-movement networks.

5) Fig 4: What is the intuition for why congruence (between the structure of latent trajectories and motor output) during *preparation* is important, since preparation precedes the actual movement? This begs the questions of how preparation activity relates to execution activity in this network.

6) Fig 5: Can the authors propose how to measure whether there is congruent structure in real data? If so, can the authors measure the congruency of the activity in different animals (by digging up old data) and using that to predict how fast each animal adapts?

7) The use of the word "congruency" was confusing. On L237-238, the definition of congruency was based on the relationship between the latent trajectories and the motor output. However, in Fig 5, the definition of congruency did not need to refer to the motor output. Can this be clarified?

In the context of Fig 6, congruency was used to refer to the relationship between the latent trajectories and the perturbation. Is this a second type of congruency?

Minor comments:

- Fig 2E inset: is it expected that the real data has higher variance?
- Clarify what is being shown in Fig 4E
- Fig 4L: is this preparation or execution activity?
- L315-316: re: "small changes in network weights", is this discussed in the main text at all?
- Equation 0.4: Is the preparation period included here? What is $t=50$ relative to the target signal or go cue?
- The authors use the term "repertoire" to refer to both the neural activity and to the movements. This can be confusing.

Response to reviewer comments

Reviewer #1 (Remarks to the Author):

The authors use a recurrent neural network model to explore how prior training/learning (“de novo motor learning”) may influence subsequent learning dynamics (“adaptation”). They do so by training naïve RNN models on different initial motor tasks and then seeing how that initial training influences the model’s ability to adapt to a perturbation of this task. This setup allows the authors to systematically control for training experience (e.g., using the exact same initial model and only changing the training history), which cannot be readily done in experiments. Through analyses and model variations, the authors find that the initial task training influences adaptation and candidate sources of this influence: changes in the number of neural activity patterns the model generates and the structure of these activity patterns. One of the more compelling and novel findings in the study is a simple demonstration that how neural activity patterns are structured in relation to input cues influences how the model can (or cannot) adapt to future perturbations. The study brings computational RNN methods to bear on questions about how population neural activity shapes motor learning, which has primarily been explored experimentally. The study therefore provides new perspectives on existing studies in motor and brain-computer interface learning and will be of interest to those communities.

Overall, I find the manuscript to be clearly written, the work thorough and sufficient to back up the primary claims being made. I do have a few comments and suggestions for areas to improve the paper and some technical questions/points of confusion that should be addressed before the paper is suitable for publication.

We thank the reviewer for their enthusiasm about our work and its relevance to the larger learning community. We appreciate their recommendations which we feel helped improve the paper.

1. Some phrasing in the introduction and abstract should be refined to reflect the current literature more accurately with regard to the relationship between network connectivity and neural activity patterns. Much of the manuscript’s motivation focuses on a line of research investigating “neural manifolds”. While this literature often emphasizes the interpretation that neural manifolds reflect the underlying anatomical connections within a network of neurons, these studies almost exclusively use correlations, which is a measure of functional connectivity. The relationship between structural and functional connectivity metrics is a significant outstanding challenge in neuroscience (e.g., Das & Fiete, Nature Neuroscience 2020). The authors must improve the care in how they discuss this work and avoid conflating functional and structural connectivity. For example, the authors cite Sadtler et al. Nature 2014 and Oby et al. PNAS 2019 as providing evidence that “this neural manifold is likely shaped, at least partly, by the underlying connectivity of the network”. Those studies show that neural correlation structures influence learning behavior – they do not prove a causal relationship between the neural manifold and the underlying structural network. Similarly, the authors later cite the Oby paper to defend the statement that generating new activity patterns necessitates changing synaptic connectivity – the paper does not directly prove this point. Indeed, past RNN modeling work highlights that BCI learning dynamics may not be purely reflect the degree of synaptic weight changes required (Feulner & Clopath, PLoS Comp Biol 2021), highlighting the complexity of neural activity pattern – structural connectivity relationships.

Thank you for highlighting this. We have edited the text in the Introduction to better reflect the current literature, especially delineating studies that have used causal manipulations (Okun et al 2014, Marshel et al 2019) or comparisons between sleep and awake behavior states (Gardner et al 2022) to provide direct evidence that neural manifolds reflect circuit constraints. We have also highlighted studies that provide post hoc support for this notion by probing learning (Sadtler et al 2014, Oby et al 2019, Golub et al 2018) or adaptation to sensorimotor

perturbations (Perich et al 2018, Sun et al 2022). We have also highlighted previous modeling studies looking at the potential mechanisms underlying those effects (Feulner and Clopath 2021, Feulner et al 2022).

2. I am rather unclear on how the variance calculations for figure 2 were performed. My largest source of confusion is that your potent and null space variances do not appear to add up to your total variance according to all the panels shown in 2c. I could not piece together from the methods how this would be the case. Please more fully explain the calculation methods. I would also recommend more precision in the language used in this section of the manuscript since “variance” can be calculated so many ways and has many different potential interpretations (it wasn’t clear to me until going all the way to the methods that this was trial-to-trial variance, for instance).

We apologize for the confusion. We were calculating the variance across trials per timestep and per feature (i.e. per unit for unit activity, latent dimension for latent activity, and output dimension for motor output), and plotting the median variance across all timesteps and features. This does not necessitate the potent and null space variances to add up, since they only need to add up when calculating the total variance summed over all features for each timestep.

We have now recalculated the variance using the total variance instead, and we have clarified this in the main text (Fig. R1i-m). Crucially, the general trends remain the same across both types of variance calculations, although the increase in output variance for larger repertoires is not as strong for the total variance. This is likely due to two things. First, when variance is summed across output features, the variance in the x direction dominates over the variance in the y direction; single-movement repertoire networks have notably less variance in the y direction, so the difference in variance between single-movement and multi-movement repertoire networks would be less pronounced. This first effect should not be enough to significantly diminish the trend, however, since the trend remains strong for networks trained on synthetic data (Fig. R1a,b,f-h). This suggests that variance in the experimental data used for training may be strong enough to mask the differences in variance between single-movement and multi-movement networks. The combination of these two effects lead to a weaker trend in the output variance, but the results based on the synthetic data show that the trend still exists.

We have now added this alternative way of calculating variance (inter-trial variability) to the paper as the new Supplementary Figure S3 and explained it appropriately.

Figure R1. Trends in variance remain robust for synthetic movements and different measures of variance. a-h. Networks were trained to produce motor output (**b**) based on simulated (**a**) rather than actual "hand trajectories" based on monkey movements. Simulated trajectories did not have trial-to-trial variability for a given movement. **c-e.** Same as Fig. 2c-d in the main text but for networks trained on simulated hand trajectories. **f-g.** Same as (**c-e**) but for total variance summed across all features (across all units for unit activity, latent dimensions for latent activity, and output dimensions for position) rather than the variance for each individual feature. Variance is still calculated for each timestep across trials. Note that the patterns remained the same for synthetic trajectories as when trained on actual hand trajectories, showing that constrained dynamics are not a byproduct of variability in the monkey movements they were trained on. **i-m.** Networks were trained to produce motor output (**i**) based on actual "hand trajectories" based on monkey movements (**j**). These are the same simulations as in the main text. **k-m.** Same as (**f-g**) but for networks trained on experimental trajectories. Note that the trends remain the

same. While some trends are not as strong, they are likely due to masking from the variability inherent in experimental trajectories, since the trends remain strong for the networks trained on synthetic trajectories.

3. Given that your different models don't generate the same overall behavior (e.g. figure 2E and figure S2), I think it would be beneficial to present an analysis showing that this is not purely an artifact of things like training duration. The loss curves suggest all the models have converged similarly, but spelling out that you see this consistent difference independent of total training would make the point more convincing.

To demonstrate the robustness of our results, we have rerun the major simulations while varying many hyperparameters, including the number of training trials. All of the trends in our main results have generally remained robust to these changes (Fig. R2).

We have added this extensive validation as the new Supplementary Figure S12.

Figure R2. Main results remain robust to changes in modeling parameters. Parameters in the neural network model were varied one at a time and simulations were run with these changes. The parameter values in red denote the original value used in the simulations in the main text. The red boxed plots denote the results for the simulations in the main text. The values in orange in (a) denote the changed values. The trends in the results were generally robust to changes in parameters. **a.** Loss curves during skill learning for networks with angular inputs. Originally presented in Fig 3b in the main text. **b.** Loss curves during skill learning for networks with categorical inputs. **c.** Variance in unit activity for angular input networks. Originally presented in Fig 3c in the main text. **d.** Decay constants for angular and categorical input networks. Originally presented in Fig 4i in the main text. **e.** Decay constants for angular and categorical input networks. Originally presented in Fig 4l in the main text. **f.** Output for reassociation perturbation for angular input networks. Decay constants for angular and categorical input networks. Originally presented in Fig 6c in the main text. **g.** Output for reassociation perturbation for angular input networks. Decay constants for angular and categorical input networks. Originally presented in Fig 6f in the main text.

4. I am not sure that the paper benefits from using the term “de novo learning”, since ultimately the question being tested is whether past learning/experience influences future learning. I understand the utility of using existing terms in the motor literature and they are broadly used in accordance with the literature (acquiring a new skill being thought of as different from adapting an existing skill). But is the fact that the RNN is learning reaches for the first time the real point being made? The manuscript does not perform tests to somehow distinguish de novo learning from general prior training. I would suggest that it might be more appropriate to restrict the claims to initial or prior training/learning influencing subsequent adaptation. The potential relationship to concepts of de novo learning seems like a useful and interesting discussion point.

We thank the reviewer for their suggestion, which we think raises an interesting point. However, upon reflection, we have decided to keep the term *de novo* learning since, as the reviewer points out, we want to highlight the distinction between learning a new skill and adapting an existing one. We feel that the potential alternative of initial/prior learning does not fully capture this dichotomy. We do agree that the distinction between initial training and *de novo* learning merits further investigation, however. In future work, we plan to investigate how networks that are pretrained on many tasks learn to perform a new skill (*de novo*), compared to networks trained from scratch on this same new skill.

Minor/specific points:

1. Structurally, I find it a bit odd that the manuscript includes a conceptual summary figure (Fig. 5) before they have finished making all the results-based claims. Figure 6 is important for making their claim about “congruent structure”, so having that claim in figure 5 seems like the cart before the horse.

While working on the revised paper we have come to the conclusion that the conceptual summary figure is no longer necessary and *we have removed it from the manuscript*.

2. In the text, being sure to clarify that “congruent structure” is with respect to the perturbation would be helpful – this isn’t always clear.

We apologize for the confusion. The reviewer is correct that the definition of congruency was unclear in the text. *We have edited the manuscript for clarification*.

3. What is the dashed line figure panels 6B and 6E?

The line was included to facilitate comparison between the loss curves between the angular and categorical input networks. *We have added a clarification in the figure caption*.

4. It’s not entirely clear to me how panel 6D supports the statement that “neighboring trajectories interfered with one another during adaptation such that the adapted motor output become overlapped”. Overlap is not very visually obvious in D (the neural space).

In this statement, we are specifying that the *motor output* (Figure 6C) rather than the latent trajectories in the neural space were overlapped. In Figure 6C, you can see that the motor outputs are no longer distinct between the different movements, and thus they are overlapping.

5. While the conceptual labels above groups of panels is clever, they occasionally cause some confusion/challenges. Figure 4K is conceptually related to panels I-L but is given its own distinct label, presumably in part because of its physical location.

We are glad the reviewer likes our conceptual labels. Figure 4K is indeed related to panels I-L but it is also distinct since it is a graphical representation of the measure in panel L, rather than actual data we are presenting. We have adjusted the panel so that its relationship to panels I-L is clearer.

6. I do not understand why Figure 2E needs an inset – can't the experimental data point be put on the same axes as the main figure? You could use a discontinuous y-axis if the scale of the data would obscure details of the model data.

We have updated the figure to reflect these changes.

Reviewer #2 (Remarks to the Author):

Summary

Animals can adapt their movements to changes in their bodies or environments, but it is an open question why some adaptations are easier than others. This is a profound question with wide implications. Here, the authors take up a longstanding hypothesis – that the history of learned movements (one’s ‘repertoire’) constrains what is learnable – and address it by analyzing a neural network with a known learning history. Applying many of the same analyses applied to animals to this neural network, the authors demonstrate a range of empirical phenomena, some of which match experimental findings (such that adaptations are easier when they align with the dimensions of largest variance in neural space) and some that are new predictions. A central claim is that having wider repertoires facilitates adaptation but only when the required adaptation is ‘congruent’ with the training history (i.e. sharing an abstract task structure).

Overall, this work tackles an important problem with conceptual originality. It is a topic I find deeply interesting. However, I was left feeling like I did not yet gain a deeper theoretic or mechanistic understanding of how and why the repertoire affects adaptation ability. Without a mechanistic understanding, it is left unproved how general these results are. This could likely be addressed with further analyses and especially by linking to literature in machine learning, as I describe below.

We thank the reviewer for their enthusiasm about our work. We trust that the changes that we have made based on the comments by all three reviewers have furthered the mechanistic interpretation of our findings.

Strengths

I want to commend the authors for an excellently written manuscript and their choice of question. The introduction was a pleasure to read. The citations were all chosen carefully, and they are used well to position this paper within recent literature. Many very interesting subfields are brought together in this work.

The preparation of figures was done thoroughly, as well. Although other settings should also be analyzed, the figures and supplemental figures represented a thorough documentation of the chosen network.

We thank the reviewer for their kind words about our work.

Weaknesses

Where this study does not yet feel complete is in 1) helping to understand **why** these phenomena are observed, and 2) how general are these phenomena to neural networks in other settings. (These are of course related, as an understanding of why this happens would imply its generality).

Neural networks are a powerful model system for this set of questions not only because they recapitulate biological phenomena, but also because they (potentially) allow a deeper understanding as to the underlying mechanisms. This manuscript largely stops at empirical characterization. But why do neural networks find it easier to adapt to movements that interpolate their repertoire? Why does the output variance increase for networks with more diverse training sets? Why does noise robustness increase for such networks, as well? Why is congruence important? This paper provides an interesting set of hypotheses, but it feels as if these need to be verified in future work.

Thanks to the reviewer for the feedback. We have made several changes to the manuscript to provide more simulations, manipulations and analyses for why the results were observed. We detail those changes in the point-by-point responses below.

But why do neural networks find it easier to adapt to movements that interpolate their repertoire?

This is an interesting question. In the original paper, we partially explained this by noting that networks that learned multiple movements create intermediate states that can be used during adaptation. We supported this by showing that these networks can produce intermediate movements when prompted with intermediate inputs that they have not explicitly learned (Fig. 2i, Fig. S5b). Also, when we applied noise to the activity when networks were producing the first movement, the output moved towards the output of the second movement, suggesting that there was interpolation between the movements in both neural space and motor output (Fig. 2g). To make a more direct connection between the neural space and motor output, we have now plotted both the latent trajectories and motor output when noise was applied to the activity (Fig. R3a,b,d,e). Indeed, noise pushed networks to intermediate states in neural space that corresponded to intermediate motor output. That networks were able to access these intermediate states with noise added suggests that these states are close to the states associated with the original movement. If we think about sensorimotor adaptation as the ability to change input-output mappings (e.g. remapping a 10 degree signal to a 20 degree output during a visuomotor rotation), this analysis shows that these intermediate states necessary for the remapping are easily accessible to multi-movement networks, such that they can easily reuse them for adaptation.

We have added these analyses as the new Supplementary Figure S4.

Figure R3. Addition of noise shows underlying structure in multi-movement networks. Noise was applied to networks trained on synthetic trajectories without trial-to-trial variability to guarantee that differences would be due to different repertoires rather than different variability in the training data. **a.** Latent trajectories for an example single-movement network. Purple line, trial-averaged trajectory without increased noise added. Pink gradient and gray lines, trajectories for example trials when increased noise ($\eta = 1$) is added. Color gradient, time-course of the trial. **b.** Same as **(a)** but for an example two-movement network. Noise is only added to the first movement. **c.** 90th-percentile tangling in the neural space (Russo et al. 2018), which quantifies how deterministic future states are from current states. **d-f.** Same as **(a-c)** but for the motor output and output space. **g.** Mean-squared error (MSE) of output when noise of increasing magnitude is added to the neural activity. Line and shaded area, median and 95% confidence interval. Left: MSE for both x and y directions. Middle: MSE only for the x direction. Right: MSE only for the y direction. **h.** Same as **(g)** but for the variance in position.

*Why does the output variance increase for networks with more diverse training sets?
Why does noise robustness increase for such networks, as well?*

These are interesting questions that relate to each other. In the main results, we showed that networks with more diverse training sets have intermediate states in between the latent trajectories corresponding to learned movements (Fig. 2i, Fig.S4b-c). To probe how this leads to more variance and robustness, we examined networks that were

trained on synthetic movements without trial-to-trial variability, such that any variability in the output would be solely due to different learned repertoires.

When we applied noise to the activity of multi-movement networks, the latent trajectories were pushed towards these intermediate states and continued more forward trajectories, leading to intermediate motor output as well (Fig. R3b, Fig. R3e). In contrast, when we applied noise to the activity of single-movement networks, the latent trajectories did not have intermediate states to go to and became more twisted, leading to more twisted output (Fig. R3a, Fig. R3d). We quantified these differences in neural population trajectories by measuring the ‘tangling’ (Russo et al., 2018) in both the latent and output space, which quantifies how well future states can be predicted from current states (Fig. R3c, Fig. R3f). Single-movement networks had more tangling in both spaces, which is predictive of lower robustness: when a very ‘jumbled’ motor output rather than the expected forward output was produced, error accumulated at each timestep, leading to higher MSE (Fig. R3g). Since the output was more jumbled, it was also restricted to a smaller range, leading to lower variance (Fig. R3h). Specifically, multi-movement networks had much more variance in the y direction since there was more variance in the y direction for movements in multi-movement repertoires (Fig. R3h). As multi-movement networks were pushed into intermediate states, they then mostly deviated in the y direction. This led to greater variance in general for multi-movement networks.

We have added these analyses as the new Supplementary Figure S4.

There is a large field of learning theory in neural networks that could help explain these results, but this field is not referenced. The authors are missing a powerful opportunity to leverage such deep learning theory. I would be excited to a review a revision which makes such an attempt.

Relatedly, the current analyses focus rather narrowly on a single setting of just 4 reaching movements, but make rather general conclusions about adaptation (and transfer learning) in neural networks in the abstract. If this is a general truth, it would be of wide interest in the machine learning community as well. However, is not yet clear how far the current conclusions will generalize to other networks or settings, and especially to more complicated movement regimes and networks. Without proven generality it is also hard to extrapolate these results to biological networks.

The generality of these many findings could be proved by empirically evaluating a range of other networks in more complicated movement (or non-movement) settings. (I would especially consider benchmarks in the ‘imitation learning’ field, which matches this current setting.) Alternatively, generality could be provided by a theoretical understanding of why the current network behaves as it does. (I’ve provided some suggestions below).

We thank the reviewer for their suggestions. We agree that we could strengthen our findings by examining networks in more complicated movement settings. To address this, we trained networks on a more complicated task where they had to output elliptical movements with various geometries by producing different sinusoids (Fig R4). We chose this task because the oscillatory behavior requires more complicated timing and dynamics, and this type of task is reminiscent of many studies in motor control (Schwartz 1993, Schwartz 1994, Marshall et al. 2023). In our setup, networks generated (two-dimensional) elliptical movements with various geometries by producing two sinusoids: a cosine wave and a sine wave representing x and y positions, respectively (Fig. R4a).

We manipulated this new task to test the generalizability of our previous findings based on the standard center-out reaching task. First, we previously showed that learning multiple movements is necessary to create intermediate

states that facilitate adaptation. To test this, we trained the networks on two sets of repertoires with different numbers of movements (Fig. R4b). In one set of repertoires, networks learned one to four movements consisting of cosine waves of different amplitudes, but of sine waves of the same amplitude, creating ellipses of different elongations in the x direction. The other set of repertoires was similar, but with sine waves of different amplitudes and cosine waves of the same amplitude, creating ellipses of different elongations in the y direction. Second, we previously showed that the structure between the inputs, neural space, and the changes imposed by the perturbation all have to be congruent to facilitate adaptation. To test the effects of different types of inputs, we encoded the target signal of the sine wave as a continuous input representing the amplitude and the target signal of the cosine wave as a categorical one-hot encoded input (Fig. R4a). To test the effects of different perturbations, we asked all networks to adapt to 1) a ‘sine wave perturbation’ where they had to produce a movement with a larger sine wave amplitude, 2) a ‘cosine wave perturbation’ where they had to produce a movement with a larger cosine wave amplitude, or 3) a ‘reassociation perturbation’ where they had to reassociate target cues to different movements (Fig. R4b).

We could make several predictions based on our previous findings (Fig. R4b). Regarding the networks that learned varied sine waves, we first predicted that networks with larger repertoires would adapt faster to the ‘sine wave perturbation’ since the continuous amplitude input encoding of the sine wave is congruent to the continuous amplitude change required by the perturbation. Second, these networks would not adapt faster to the ‘cosine wave perturbation’ since they have not learned multiple cosine waves and thus do not have the intermediate states necessary to facilitate adaptation. Third, these networks would not adapt well to the ‘reassociation perturbation’ since their continuous inputs are incongruent to the categorical changes required by the perturbation. Regarding the networks that learned varied cosine waves, we first predicted that they would not adapt faster to the ‘sine wave perturbation’ since they have not learned multiple sine waves. Second, networks with multi-movement repertoires would adapt faster to the ‘cosine wave perturbation’ than those with single-movement repertoires since they have some structure. However, larger multi-movement repertoire networks would not adapt faster since the categorical input encoding of the cosine wave is incongruent to the continuous change required. Third, these networks would adapt well to the ‘reassociation perturbation’ since their categorical inputs are congruent to the categorical changes required.

Indeed, the results support all of our predictions (Fig. R4c-g), showing that our findings generalize to more complicated movement settings. Specifically, it demonstrates that different components of a given movement—here, the x and y components—can be learned using different input cues, leading to contrasting adaptations to different perturbations.

We have added these new simulations and results confirming our previous findings as a new section and main figure in the revised version of the manuscript.

Figure R4. De novo learning and adaptation for a more complex sinusoidal task. **a.** Networks were trained to produce cosine and sine waves representing x and y positions of different ellipses, respectively. Networks produced one of up to four possible sine and cosine waves each, of different amplitudes. Networks were given a hold signal that indicated movement initiation, a continuous sine wave target signal that indicated the amplitude of the sine wave, and a categorical 'one-hot-

encoded' cosine wave target signal. **b.** Networks were trained on repertoires with different numbers of movements (from one to four) to model *de novo* learning. Within each repertoire, the movements could consist of either different sine waves and a constant cosine wave ('varied sine wave'), or different cosine waves and a constant sine wave ('varied cosine waves'). Following *de novo* learning, networks were separately trained to counteract to 1) a sine wave perturbation where they had to produce the one common movement with a larger amplitude for the sine wave, 2) a cosine wave perturbation where they had to produce the common movement with a larger amplitude for a cosine wave, and 3) a reassociation perturbation where they had to produce different movements given learned target cues. Given previous results, varied sine wave networks with larger repertoires were predicted to adapt faster to the sine wave perturbation since sine waves were indicated with continuous amplitude inputs that are congruent with the sine wave amplitude perturbation. Varied cosine wave networks were predicted to adapt better to the reassociation perturbation since cosine waves were indicated with categorical inputs that are congruent with the categorical reassociation perturbation. **c.** Cosine waves (x position), sine waves (y position), and elliptical output produced by networks trained on each repertoire with varied sine waves after *de novo* learning and each of the perturbations. **d.** Loss during *de novo* learning and adaptation to each of the perturbations. Loss was calculated as the mean-squared error between the network output and target positions. Line and shaded surfaces, smoothed mean and 95% confidence interval across networks of different seeds. **e-f.** Same as (c-d) but for networks trained on repertoires with varied cosine waves. For the reassociation perturbation, 45% of networks trained on either 3- or 4-movement repertoires with varied sine waves were able to adapt (MSE < 0.4), compared to 70% for those with varied cosine waves. **g.** Decay constants for exponential curves fitted to the loss curves in (d) and (f) for the sine and cosine wave perturbations.

To summarize, this manuscript implies bold claims about transfer learning in neural networks, but does not utilize the extensive literature on this question in machine learning. This has the effect of leaving uncertain why these phenomena are observed or in what other settings they might be expected to occur.

We agree that examining transfer learning through the lens of machine learning is interesting, and we are indeed exploring transfer learning in another ongoing study. However, we feel that the current paper is complex enough without addressing this issue directly. Rather, we show through additional simulations and analyses why these phenomena are observed and how they might generalize to additional settings.

Questions

ll 203-215: That distances in neural space are proportional to distances in motor output would seem to be guaranteed by the use of a linear readout. Differences in proportionality in different settings would then reflect differences in the projection of activity upon the null/potent space of the readout. Would the authors predict that adaptation ability is a consequence of the fraction of preparation/execution activity that projects into the potent space? If preparatory activity is by definition in the null space, this would seem to imply that adaption ability follows having a small variance in preparatory activity? Just trying to follow the logic.

It seems that your question stems from our original explanation that angular input networks adapt faster because the structure in their neural space is congruent to the structure in the motor output. Since the structure in motor output has to be congruent to the structure in the potent space, it seems that the degree of congruence (and thus the adaptation ability) would then be dependent on the activity in the null space or during preparation. However, our comparison between the results of the visuomotor rotation perturbation and the reassociation perturbation revealed that it is actually important for the structure—relative organization of the activity patterns underlying each movement—in neural space to be congruent to structure of the perturbation and how it acts on the motor output.

The structure of the motor output is of course important for this, but it is not the structure of the motor output alone that determines the ease of adaptation. Based on this and other comments by the reviewers, we realized that this distinction is rather unclear in the original manuscript, and we have edited the manuscript for clarification. In particular, we now highlight that the learning experience by which a certain repertoire is acquired, which we could manipulate by presenting inputs that had different relationships between them (one-hot vs continuous), plays a crucial role in shaping the structure in activity space and facilitating or hindering subsequent learning. Thus, although it is still important to compare the structure of the neural space and the structure of the motor output in Lines 203-215 since it allows us to make the prediction that changes in the latent trajectories will lead to similar changes in the motor output—i.e. intermediate latent trajectories would produce intermediate motor output—, this is not the main explanation for the adaptation results.

We have revised the text extensively to highlight the role of how the relationship among the inputs shapes structure in neural activity space, and how this structure facilitates or hinders adaptation depending on whether it is congruent with the changes required by the perturbation.

ll 227-238: I'm not yet convinced that this categorical encoding task **proves** adaptation is a consequence of the congruence between neural space and motor output. The categorical encoding task could negatively affect angular adaption for other reasons than the congruence. This seems to be a correlation taken as a causal mechanism. (But why stop there? In neural networks we can know the mechanism and analytically describe learning dynamics.)

We understand the reviewer's concern about sufficient evidence for the role of congruence in adaptation. To strengthen this relationship, we used the new task above (Fig R4) to extend our results about the importance of congruence to new ways of manipulating the inputs and new types of perturbations. We saw that the results still hold with this new task: networks that learned varied sine waves with amplitudinal inputs adapted faster to perturbations that required amplitude changes, and networks that learned varied cosine waves with categorical inputs adapted better to perturbations that required categorical changes (Fig R4d–g).

Further, we came up with a metric to quantify congruence (see Methods or the description in the response below) and applied it to experimental data of monkeys adapting to a visuomotor rotation during a center out task. Across two monkeys, the rate of adaptation was correlated with the degree of congruence (Fig R5), providing experimental support for our predictions.

In addition to adding the new section around Figure R4 to the revised version of the manuscript, we have also added the new Figure R5 as part of Supplementary Figure S8.

Figure R5. Congruency correlates with rate of adaptation to a visuomotor rotation perturbation in experimental monkey data. Monkeys adapted to a visuomotor rotation perturbation during a center-out reach task (3 sessions for Monkey C, 3

sessions for Monkey M, 10 sessions for Monkey M2; Monkey M and M2 were the same monkey, but they are treated separately since the recordings were from different regions in motor cortex). Monkeys were assumed to be given angular inputs since they were shown the location of the target centered around a circle. Congruency was calculated between the structure of angular inputs and the structure of their neural spaces. Error curves were calculated for each session based on the angular error of monkey reaches in the first 150 ms. The rate of adaptation was measured as the decay constants for exponential curves fitted to these error curves.

ll 275-305: Could the authors provide a more precise definition of ‘congruence’ between the type of perturbation and the neural structure? Is it possible to define this mathematically? This would help the reader to understand when else to expect adaptation to be easy/hard.

We agree that having a more precise definition for ‘congruence’ would be useful. While the reviewer mentioned having a definition for the congruence between the structure of the neural space and the perturbation, we would like to note that there must be congruence between three levels of structure: the inputs, the neural space, and the perturbation. Following the reviewer’s suggestion, we tried to come up with a metric that mathematically defines the congruence at all three levels.

We defined congruence as the similarity in the relative organization of elements from two groups. For the structure of the inputs, we constructed a representational dissimilarity matrix (RDM) (Kriegeskorte et al 2008) by calculating the cosine dissimilarity between pairs of target signal input vectors associated with each movement. For the structure of activity in neural space, we constructed a RDM by calculating the distances between the latent trajectories associated with each movement. We can then measure the congruence between the relative organization of the inputs and activity in neural space structures by simply calculating the Pearson’s correlation coefficient between the upper triangular regions of their respective RDMs. With this measure, angular inputs (categorical inputs) have a high degree of congruency with the latent trajectories of angular input (categorical input) networks, and a lesser degree of congruency with the opposite set of latent trajectories (Fig R6).

Figure R6. Congruency between input and neural structure. Networks were given either angular (a) or categorical (b) inputs. After training on repertoires of one to four movements, they then had to adapt to a counterclockwise VR perturbation of 10° . c. Left: Input structure, measured as the representational dissimilarity matrix (RDM) of cosine dissimilarities between target

signal input vectors for pairs of movements for an example network with angular inputs. Right: Neural structure, measured as the RDM of normalized median Euclidean distances between latent trajectories during preparation and execution for different movements for the same network. **d.** Same as Panels c but for an example network with categorical inputs. **e.** Congruency between the input structure and neural structure, quantified as the Pearson's correlation between RDMs for each structure. Congruency for the opposite, mismatched input is shown as a control. Circles and error bars, means of congruency values for each seed and 95% confidence intervals with bootstrapping.

The structure of the perturbation, however, is harder to define mathematically. Qualitatively, it is easy to understand how an angular perturbation is congruent to angular inputs or how a categorical perturbation is congruent to categorical inputs. However, it is quantitatively difficult to measure this since the nature of the perturbation can vary widely. For example, we can represent the reassociation perturbation for a four-movement repertoire as a 4x4 permutation matrix since it acts on all four movements, but we cannot construct a similar matrix for the visuomotor rotation since it only acts on one movement. Thus, we can only define the structure of the permutation qualitatively.

We have added the new analysis establishing the congruence between the relative organization of the inputs and activity in neural space as part of main Figure 4, and describe the new analyses appropriately in the text.

l 495: Were the networks trained with backpropagation through time, or to minimize the objective at each timepoint (ignoring history)? I'm not getting at anything here, just curious.

Yes, we used backpropagation through time to account for the dependencies across time steps.

l 511: Again just curious, but why were firing rates smoothed with a Gaussian kernel? I don't think it would affect results either way... but I'd bet that running PCA on this timeseries and keeping only lower PCs will result in a lowpass effect anyways.

We wanted to compare the results to experimental data (Fig S6), and we generally preprocess the experimental recordings by applying a Gaussian kernel (std = 50 ms) to the binned square-root transformed firings of each unit, as is common in the field (e.g. Yu et al 2009, Gallego et al 2018). To make the results more comparable, we applied the same Gaussian kernel to the simulated firing rates. As the reviewer suspected, this preprocessing step does not affect the results of PCA since the latent trajectories are qualitatively the same with and without this step (Fig R7).

Figure R7. Gaussian smoothing does not affect latent trajectories. **a.** Reproduced from Fig. 4b in main text. Latent activity for an example network with angular inputs trained on four movements during preparation (500 ms before go cue) and execution (1000 ms after go cue). Gaussian smoothing was applied to the neural network rates before principal component analysis (PCA) was performed. Each trace corresponds to the trial-averaged activity for each movement projected on neural manifold computed before adaptation. Solid lines, activity before adaptation; dotted lines, activity after adaptation. **b.** Same as (a) but Gaussian smoothing was not applied to the neural networks rates before PCA.

Suggestions

Some papers the authors may wish to cite are,

Krakauer, J. W., Mazzoni, P., Ghazizadeh, A., Ravindran, R., & Shadmehr, R. (2006). Generalization of motor learning depends on the history of prior action. *PLoS biology*, 4(10), e316.

Sing, Gary C., et al. "Primitives for motor adaptation reflect correlated neural tuning to position and velocity." *Neuron* 64.4 (2009): 575-589.

In Sing et al. (2009), the authors model motor output as a linear combination of the motor repertoire (the 'motor primitives'). (It is interesting to note that this framework resembles some findings in the current work, such as Fig S4, that perturbations can be adapted to if the movement is within the range spanned by learned examples.)

Thank you for these suggestions. *We have added the relevant citations in the appropriate places of the manuscript.*

If I may, I'd like to suggest one manner in which the authors might show mathematically that adaptation ability in neural networks relates to the structure of neural activity, and in turn to the history of trained movements. Take or it leave it; there are many ways in which the generality of this paper's claims beyond the single 4-movement setting could be proved.

In modern machine learning parlance, the model of Sing et al. would be called a (linear) kernel machine. What is interesting and powerful about this connection from perspective of adaption is that the learning dynamics of kernel machines can be described exactly. Thus, one could fairly straightforwardly prove what sorts of adaptations are easy/hard to learn. Essentially, the easy perturbations to learn are ones that map onto the largest PCs of the last

neural population before the linear combination (the ‘kernel’). If this direction interests you, I would recommend the following publications:

Bordelon, B., & Pehlevan, C. (2022). Population codes enable learning from few examples by shaping inductive bias. *Elife*, 11, e78606.

Flesch, T., Saxe, A., & Summerfield, C. (2023). Continual task learning in natural and artificial agents. *Trends in Neurosciences*, 46(3), 199-210.

Additionally, I recommend the authors consult the following publication, which describes analytically how the Sadler et al. result (that BCI tasks are easier when they are aligned with the structure of the learning task) can arise from gradient descent learning.

Humphreys, P. C., Daie, K., Svoboda, K., Botvinick, M., & Lillicrap, T. P. (2022). BCI learning phenomena can be explained by gradient-based optimization. *bioRxiv*, 2022-12.

Thank you for your suggestion. We agree that the present paper is only the first step in an exciting—at least for us—direction, and that there are many promising lines of inquiry ahead. Coincidentally, we are currently working on a continual learning project, and one of the aspects we are investigating is the difference in adaptation to perturbations that require learning along the first PCs versus in directions that explain less variance in population activity. Perhaps, we will in the future be able to discuss our ongoing work on continual learning with the reviewer since it seems we are thinking along the same lines? Regarding analytical derivations, while they are indeed very powerful and elegant, they become intractable for recurrent networks like the ones we have modeled, which is why most of the current work in the references above focus on feedforward networks. Nevertheless, thank you for the suggested papers; we have cited the ones that we found relevant.

Reviewer #3 (Remarks to the Author):

This paper uses a recurrent network modeling approach to study how the breadth and type of prior motor learning influences how quickly and accurately the network is able to adapt to perturbations in the motor output. The authors find that networks show faster adaptation when the network has learned a greater breadth of movement angles and when the activity is structured in the network in such a way that it is matched with the type of perturbation. Overall, this is an interesting paper on a timely topic. However, the conditions studied within the network is limited in scope and its relevance to how learning takes place in the brain is currently unclear (see details below).

We thank the reviewer for their interest in our work, and trust that the new analyses and simulations that we have now added have made them more enthusiastic about its scope.

Major comments:

1) The authors consider only a center-out movement task, and only 4 targets that only span 40 degrees. And the perturbation is to only a single target. It is unclear whether such limited scope allows the broader statements the authors make about how learning influences adaptation in this network. Making a link to the brain is an even bigger stretch. The authors should demonstrate a wider range of learned tasks by the network (beyond center-out movements), and test how the network adapts subsequently.

We thank the reviewer for their suggestions. As stated in the original manuscript, our choice of a relatively simple task and a common perturbation was intentional because we wanted to have the simplest model to answer the question of how the motor repertoire and the process by which it is learned influence subsequent adaptation. We have now extended our findings by replicating our main findings on networks trained on a more complicated task where they had to output elliptical movements by producing different sinusoids (Fig R8). We chose this task because the oscillatory behavior requires more complicated timing and dynamics. Also, the generation of time varying forces is a well established topic in motor control, which has been tackled both in classic and recent studies (Schwartz 1993, Schwartz 1994, Marshall et al. 2023). Notably, here we are asking the networks to draw ellipses with varying geometries by producing two sinusoids: a cosine wave and a sine wave representing x and y positions, respectively.

We manipulated this new task to test the generalizability of our previous findings. First, we previously showed that learning multiple movements is necessary to create intermediate states that facilitate adaptation (Fig. R8a). To test this, we trained the networks on two sets of repertoires with different numbers of movements (Fig. R8b). In one set of repertoires, networks learned one to four movements consisting of sine waves of different amplitudes, but of cosine waves of the same amplitude, creating ellipses of different elongations in the x direction. The other set of repertoires was similar, but with cosine waves of different amplitudes and sine waves of the same amplitude. Second, we previously showed that the structure between the inputs, neural space, and perturbation have to be congruent to facilitate adaptation. To test the effects of different inputs, we encoded the target signal of the sine wave as a continuous input representing the amplitude and the target signal of the cosine wave as a categorical one-hot encoded input (Fig. R8a). To test the effects of different perturbations, we asked all networks to adapt to 1) a ‘sine wave perturbation’ where they had to produce a movement with a larger sine wave amplitude, 2) a ‘cosine wave perturbation’ where they had to produce a movement with a larger cosine wave amplitude, or 3) a ‘reassociation perturbation’ where they had to reassociate target cues to different movements (Fig. R8b).

Figure R8 (Reproduced from Fig R4). *De novo* learning and adaptation for a more complex sinusoidal task. a. Networks were trained to produce cosine and sine waves representing x and y positions of different ellipses, respectively. Networks produced one of up to four possible sine and cosine waves each, of different amplitudes. Networks were given a hold signal that indicated movement initiation, a continuous sine wave target signal that indicated the amplitude of the sine wave, and a

categorical 'one-hot-encoded' cosine wave target signal. **b.** Networks were trained on repertoires with different numbers of movements (from one to four) to model *de novo* learning. Within each repertoire, the movements could consist of either different sine waves and a constant cosine wave ('varied sine wave'), or different cosine waves and a constant sine wave ('varied cosine waves'). Following *de novo* learning, networks were separately trained to counteract to 1) a sine wave perturbation where they had to produce the one common movement with a larger amplitude for the sine wave, 2) a cosine wave perturbation where they had to produce the common movement with a larger amplitude for a cosine wave, and 3) a reassociation perturbation where they had to produce different movements given learned target cues. Given previous results, varied sine wave networks with larger repertoires were predicted to adapt faster to the sine wave perturbation since sine waves were indicated with continuous amplitude inputs that are congruent with the sine wave amplitude perturbation. Varied cosine wave networks were predicted to adapt better to the reassociation perturbation since cosine waves were indicated with categorical inputs that are congruent with the categorical reassociation perturbation. **c.** Sine waves (x position), cosine waves (y position), and elliptical output produced by networks trained on each repertoire with varied sine waves after *de novo* learning and each of the perturbations. **d.** Loss during *de novo* learning and adaptation to each of the perturbations. Loss was calculated as the mean-squared error between the network output and target positions. Line and shaded surfaces, smoothed mean and 95% confidence interval across networks of different seeds. **e-f.** Same as (c-d) but for networks trained on repertoires with varied cosine waves. For the reassociation perturbation, 45% of networks trained on either 3- or 4-movement repertoires with varied sine waves were able to adapt ($MSE < 0.4$), compared to 70% for those with varied cosine waves. **g.** Decay constants for exponential curves fitted to the loss curves in (d) and (f) for the sine and cosine wave perturbations.

We could make several predictions based on our previous findings (Fig. R8b). Regarding the networks that learned varied sine waves, we first predicted that networks with larger repertoires would adapt faster to the 'sine wave perturbation' since the continuous amplitude input encoding of the sine wave is congruent to the continuous amplitude change required by the perturbation. Second, these networks would not adapt faster to the 'cosine wave perturbation' since they have not learned multiple cosine waves and thus do not have the intermediate states necessary to facilitate adaptation. Third, these networks would not adapt well to the 'reassociation perturbation' since their continuous inputs are incongruent to the categorical changes required by the perturbation. Regarding the networks that learned varied cosine waves, we first predicted that they would not adapt faster to the 'sine wave perturbation' since they have not learned multiple sine waves. Second, networks with multi-movement repertoires would adapt faster to the 'cosine wave perturbation' than those with single-movement repertoires since they have some structure. However, larger multi-movement repertoire networks would not adapt faster since the categorical input encoding of the cosine wave is incongruent to the continuous change required. Third, these networks would adapt well to the 'reassociation perturbation' since their categorical inputs are congruent to the categorical changes required.

Indeed, the results support all of our predictions (Fig. R8c-g), showing that our findings generalize to more complicated movement settings. Specifically, it demonstrates that different components of a given movement—here, the x and y components—can be learned using different input cues, leading to contrasting adaptations to different perturbations.

We have added these new simulations and analyses that confirm our main results during a more complex task as a new section and main figure in the revised manuscript.

2) The authors should examine how their modeling assumptions in L452-458 influence their results. In particular, how did the authors choose the variances (or ranges) of each of the probability distributions from which they drew

their noise, initial states, and model parameters? How does varying those assumptions alter the scientific conclusions of the study?

The initial hyperparameters were either chosen based on existing models or chosen to allow the networks to learn the desired repertoires, but the specific hyperparameters are not particularly important for our results. To demonstrate the robustness of our results, we have rerun the major simulations while varying many hyperparameters. All of the trends in our main results have generally remained robust to these changes (Fig. R9). *We have added these simulations supporting the robustness of our results against parameter selection as new Supplementary Figure S12.*

Figure R9 (Reproduced from Fig R2). Main results remain robust to changes in modeling parameters. Parameters in the neural network model were varied one at a time and simulations were run with these changes. The parameter values in red denote the original value used in the simulations in the main text. The red boxed plots denote the results for the simulations in the main text. The values in orange in (a) denote the changed values. The trends in the results were generally robust to changes in parameters. **a.** Loss curves during skill learning for networks with angular inputs. Presented in Fig 3b in main text. **b.** Loss curves during skill learning for networks with categorical inputs. **c.** Variance in unit activity for angular input networks. Presented in Fig 3c. **d.** Decay constants for angular and categorical input networks. Presented in Fig 4i. **e.** Decay constants for angular and categorical input networks. Presented in Fig 4l. **f.** Output for reassociation perturbation for angular input networks. Decay constants for angular and categorical input networks. Presented in Fig 6c. **g.** Output for reassociation perturbation for angular input networks. Decay constants for angular and categorical input networks. Presented in Fig 6f.

3) As presented, this paper elucidates the learning properties of a recurrent network model. The authors would like to imply that this might be how the brain learns, but at present the links are weak. Real data is contained in Fig 2E and S6, but those figures are more to demonstrate that the order of magnitude of different metrics match between model and data, rather than to show evidence that their main modeling results might hold in the brain. Can the authors make a stronger case using existing experimental data that this is how the brain learns? For example, for the first main result comparing networks which have learned different numbers of movement directions, can the authors dig up data from the Miller lab in which different animals have learned different sets of tasks over the course of their lifetimes, and then compare across animals?

To test how different existing repertoires affect adaptation, we would need experimental data from animals that acquired different motor repertoires throughout their lifetimes, and test their adaptation to different perturbations. A primary reason why we wanted to study this phenomenon through recurrent neural networks is that this is very difficult to measure experimentally, especially with existing datasets. Existing datasets, including those from the Miller lab, do not have animals that learned different sets of tasks *de novo* that are tested on different perturbations. Instead, they have animals that learned different laboratory tasks that are similar to behaviors that they normally perform (e.g. reaches, wrist movements, power grips, etc) (Gallego et al, 2018, Perich et al 2018), so it seems unlikely that animals that learned these tasks were acquiring a truly new motor skill. Additionally, animals were only tested on perturbations to their reaching movements (Perich et al 2018).

In the original manuscript we showed that neural activity in monkey motor cortex was organized in a manner that resembled that of networks that learned to perform the reaches *de novo* using continuous angular inputs (Figure S6). We have complemented this comparison with a new analysis that asks whether the amount of within-day learning is related to the congruency between the inputs and activity space (details in response to comment 6 below). Remarkably, we found a strong correlation between these two measures (Figure R10), gaining experimental support for the prediction that congruence across the structure of the inputs, neural space, and changes required by the perturbation facilitates learning.

We have now added this new analysis to the paper as part of Supplementary Figure S8, and discussed it appropriately.

Figure R10. (Reproduced from Figure R5s). Congruency correlates with rate of adaptation to a visuomotor rotation perturbation in experimental monkey data. Monkeys adapted to a visuomotor rotation perturbation during a center-out reach task (3 sessions for Monkey C, 3 sessions for Monkey M, 10 sessions for Monkey M2; Monkey M and M2 were the same monkey, but they are treated separately since the recordings were from different regions in motor cortex). Monkeys were assumed to be given angular inputs since they were shown the location of the target centered around a circle. Congruency was calculated between the structure of angular inputs and the structure of their neural spaces. Error curves were calculated for each session based on the angular error of monkey reaches in the first 150 ms. The rate of adaptation was measured as the decay constants for exponential curves fitted to these error curves.

4) Fig 2F: this results seems unexpected. If multi-movement networks have more activity in output-potent subspace relative to output-null subspace compared to a single-movement network, one would expect more noise to "leak" into the movements for multi-movement networks.

To clarify, the results do not show that there is *more activity* in the *output-potent subspace* in multi-movement networks. Rather, the results show that there is *more variance* in the *output-null subspace* for the single-movement networks compared to the multi-movement networks. This suggests that activity in the output-null space is less ordered/controlled for single-movement networks, whereas the activity in the output-potent space is equally ordered/controlled. As a result, single-movement networks become less robust (Fig 2f), since adding noise to this activity pushes the activity randomly into either output-null or output-potent spaces, leading to jumbled up movements. In contrast, adding noise to multi-movement networks led to more ordered output (Fig 2g).

To examine this further, we have now performed a new series of simulations in which we added noise to networks that were trained on synthetic movements without trial-to-trial variability, such that any differences in the output would be solely due to different learned repertoires. When we applied noise to the activity of multi-movement networks, the latent trajectories were pushed towards intermediate states and continued more forward trajectories, leading to intermediate motor output as well (Fig. R11b, Fig. R11e). In contrast, when we applied noise to the activity of single-movement networks, the latent trajectories did not have intermediate states to go to and became more twisted, leading to more twisted output (Fig. R11a, Fig. R11d). We quantified these differences by measuring the 'tangling' in both the latent and output space (Russo et al., 2018), a metric that quantifies how deterministic future states are from current states (Fig. R11c, Fig. R11f). Single-movement networks had more tangling in both spaces, which led to lower robustness: when more tangled rather than forward output was produced, error accumulated at each timestep, leading to higher MSE (Fig. R11g). *We have added these analyses as the new Supplementary Figure S4.*

Figure R11 (Reproduced from Fig R3). Addition of noise shows underlying structure in multi-movement networks. Noise was applied to networks trained on synthetic trajectories without trial-to-trial variability to guarantee that differences would be due to different repertoires rather than different variability in the training data. **a.** Latent trajectories for an example single-movement network. Purple line, trial-averaged trajectory without increased noise added. Pink gradient and gray lines, trajectories for example trials when increased noise ($\eta = 1$) is added. Color gradient, time-course of the trial. **b.** Same as (a) but for an example two-movement network. Noise is only added to the first movement. **c.** 90th-percentile tangling in the neural space (Russo et al. 2018), which quantifies how deterministic future states are from current states. **d-f.** Same as (a-c) but for the motor output and output space. **g.** Mean-squared error (MSE) of output when noise of increasing magnitude is added to the neural activity. Line and shaded area, median and 95% confidence interval. Left: MSE for both x and y directions. Middle: MSE only for the x direction. Right: MSE only for the y direction. **h.** Same as (g) but for the variance in position.

5) Fig 4: What is the intuition for why congruence (between the structure of latent trajectories and motor output) during *preparation* is important, since preparation precedes the actual movement? This begs the questions of how preparation activity relates to execution activity in this network.

We have revised our original explanation that angular input networks adapt faster because the structure in their neural space is congruent to the structure in the motor output. Our comparison between the results of the visuomotor rotation perturbation and the reassociation perturbation revealed that what is actually important is for the structure in neural space to be congruent to structure of the perturbation and how it acts on the motor output.

The structure of the motor output is of course important for this, but it is not the main factor determining the ease of adaptation. *We realized that this distinction was rather unclear in the original manuscript, and we have now thoroughly edited the manuscript for clarification*, summarizing our results as: adaptation is not only crucially driven by the existing motor repertoire, but also by how this repertoire is encoded in the input-output mapping learned initially by the network, which is contingent in the learning experience itself (e.g., whether the different inputs have meaning with respect to each other) (Bengio et al 2009).

It is still important to compare the structure of the neural space and the structure of the motor output in Fig. 4 since it allows us to make the prediction that changes in the latent trajectories will lead to similar changes in the motor output—i.e. intermediate latent trajectories would produce intermediate motor output. However, this is not the main explanation for the adaptation results.

Nevertheless, to address the question of how preparation is specifically important, we note that during movement execution, the structure of the latent trajectories is congruent to the structure of the motor output by construction, since we have a linear readout from activity to output, as is commonly assumed in the field (e.g. Hennequin et al 2014, Sussilo et al 2015, Feulner & Clopath 2021). However, incongruencies between the structure of the neural space and the structure of the motor output (or perturbation) can still arise during movement preparation. In the main manuscript, we trained the networks on a task with an explicit preparatory epoch to model the instructed delayed center-out reaching task from Perich et al (Neuron, 2018). Thus, we could clearly separate the preparatory state from actual movement execution. However, animals still have organized preparatory activity even when there is not an instructed delay period (Lara et al 2018, Zimmik & Churchland 2021). We have now run an additional set of simulations where we trained the networks on the same task, but without a delay or explicit preparatory epoch (Fig. R12). When we separated the execution epoch ($t = \text{go cue to } t = 1000 \text{ ms}$) into the first 200 ms and the last 800 ms, we saw that we could still reduce the congruency between the neural space and motor output for the networks with categorical inputs at the beginning of the execution epoch (Fig. R121a-f). Importantly, all the differences in adaptation between these networks with categorical inputs and networks with continuous angular inputs remained the same as in the main manuscript (compare Fig. R12g-k to Figure 4j-m in the revised manuscript). Thus, our results do not rely on an explicit preparatory epoch, but rather depend on the structure of the latent trajectories at the beginning of a movement.

We have now added these new analyses as Supplementary Figure S7.

Figure R12. Explicit preparatory epochs are not necessary for differences in adaptation and structure. Networks were trained to produce movements without an explicit preparatory epoch. That is, networks started movement at the target cue rather than waiting for a go cue. Without an explicit preparatory epoch, networks still had preparatory-like activity in the first 200 ms after the target cue and execution-like activity for the next 800 ms after the target cue. **a-f.** Same measures as Fig 4a-f in the main text, but with the distances in neural space separated into the first 200 ms after the target cue and 200 to 1000 ms after the target cue. **g-k.** Same measures as Fig 4l.

6) Fig 5: Can the authors propose how to measure whether there is congruent structure in real data? If so, can the authors measure the congruency of the activity in different animals (by digging up old data) and using that to predict how fast each animal adapts?

We thank the reviewer for this suggestion, which was also raised by Reviewer 2. We have now come up with a metric to measure congruency in our models, which we have also been able to apply to the experimental data. Overall, the results, outlined below, provide experimental support to the predictions provided by our model.

We defined congruence as the similarity in the relative organization of elements from two groups. For the structure of the inputs, we constructed a representational dissimilarity matrix (RDM) (Kriegeskorte et al 2008) by calculating the cosine dissimilarity between pairs of target signal input vectors associated with each movement. For the structure of activity in neural space, we constructed a RDM by calculating the distances between the latent trajectories associated with each movement. We can then measure the congruence between the relative organization of the inputs and activity in neural space structures by simply calculating the Pearson's correlation coefficient

between the upper triangular regions of their respective RDMs. With this measure, angular inputs (categorical inputs) had a high degree of congruency with the latent trajectories of angular input (categorical input) networks, and a lesser degree of congruency with the latent trajectories acquired after initial training based on a different types of inputs (categorical for the case of initial training with angular inputs, and vice versa) (Fig R13). Moreover, using this congruency metric, we saw that how fast monkeys adapted to the VR perturbation on a specific session was robustly correlated with the congruency between the inputs and activity in neural space, gaining experimental support for our modeling findings (Figure R10). *We have included this new metric in the paper as part of updated Figure 4, and the analysis of the monkey data is included within Supplementary Figure S8.*

The structure of the perturbation, however, is harder to define mathematically. Qualitatively, it feels intuitive that an angular perturbation is congruent to angular inputs or that a categorical perturbation is congruent to categorical inputs. However, it is quantitatively difficult to measure the perturbation since the nature of the perturbation can vary widely. For example, we can represent the reassociation perturbation for a four-movement repertoire as a 4x4 permutation matrix since it acts on all four movements, but we cannot construct a similar matrix for the visuomotor rotation since it only acts on one movement. Thus, we can only define the structure of the permutation qualitatively.

Figure R13 (Reproduced from Figure R6). Congruency between input and neural structure. Networks were given either angular (a) or categorical (b) inputs. After training on repertoires of one to four movements, they then had to adapt to a counterclockwise VR perturbation of 10° . c. Left: Input structure, measured as the representational dissimilarity matrix (RDM) of cosine dissimilarities between target signal input vectors for pairs of movements for an example network with angular inputs. Right: Neural structure, measured as the RDM of normalized median Euclidean distances between latent trajectories during preparation and execution for different movements for the same network. d. Same as Panels c but for an example network with categorical inputs. e. Congruency between the input structure and neural structure, quantified as the Pearson's correlation between RDMs for each structure. Congruency for the opposite, mismatched input is shown as a control. Circles and error bars, means of congruency values for each seed and 95% confidence intervals with bootstrapping.

7) The use of the word "congruency" was confusing. On L237-238, the definition of congruency was based on the relationship between the latent trajectories and the motor output. However, in Fig 5, the definition of congruency did not need to refer to the motor output. Can this be clarified?

In the context of Fig 6, congruency was used to refer to the relationship between the latent trajectories and the perturbation. Is this a second type of congruency?

We apologize for the confusion. The reviewer is correct that the definition of congruency was unclear in the text. Specifically, it is important for there to be congruency across three structures: the inputs, latent trajectories, and perturbation. Please see the reply for point 5) for further clarification. *We have thoroughly edited the manuscript for clarification.*

Minor comments:

- Fig 2E inset: is it expected that the real data has higher variance?

Yes, because the general trend we see is that the variance increases with larger repertoires. Our networks have much smaller repertoires (only one to four reaches) compared to monkeys used in the real data. Since the monkeys have much larger motor repertoires based on an entire lifetime of learning, we expected the variance in their movements to increase as well.

- Clarify what is being shown in Fig 4E

We have clarified this in the figure caption.

- Fig 4L: is this preparation or execution activity?

This measure includes data from timepoints in both preparation and execution activity. *In the caption, we added that the deviation angles were calculated "across all timesteps".*

- L315-316: re: "small changes in network weights", is this discussed in the main text at all?

Yes, this was briefly mentioned in L301-303 in the original manuscript, but it was not emphasized. *We have removed it from the discussion for clarity since it is not a main result and is not a necessary part.*

- Equation 0.4: Is the preparation period included here? What is $t=50$ relative to the target signal or go cue?

Yes, this includes the preparation period. The target cues are between 1.0-2.5s and the go cues are between 2.5-3.0s, so $t=50$ (500 ms) is 0.5-2.0s or 2.0-2.5s before the target or go cues, respectively. The timings are indicated under "Task" in the Methods.

- The authors use the term "repertoire" to refer to both the neural activity and to the movements. This can be confusing.

We have edited the text to avoid confusion.

REVIEWERS' COMMENTS

Reviewer #1 (Remarks to the Author):

I thank the authors for their very thorough revisions which, in my opinion, have very significantly strengthened the flow of the arguments and overall strength of their claims. The revisions have largely addressed my main concerns with one exception:

In my initial review I highlighted that the link between what their model is testing and the concept of "de novo learning" as described in the motor learning literature, is not necessarily a straight forward one. Since it's not completely straight forward, I don't fundamentally object to them retaining the framing within the manuscript but think the manuscript shouldn't be published without including a discussion of these nuances and potential limits on how their model maps to experiments and existing literature/concepts around de novo learning.

Reviewer #2 (Remarks to the Author):

Thank you for the comprehensive responses. The changes made to the manuscript have addressed some of my concerns, but not all. In particular, I continue to find it unsatisfying that it is not analyzed why (or mechanistically how) these neural network models find it easier to adapt after certain training regimens. In my opinion the time has passed where it is sufficient to present a simple comparison of brains and ANNs as an explanation of a brain phenomenon. It is time, and indeed very possible, to look deeper into the reasons why these phenomena occur.

Seeing as it is difficult to use theory to answer this question, it would still be satisfying to see analyses testing hypotheses about how, mechanistically speaking, the repertoire affects future learning. At a minimum, I would like to see a discussion paragraph asking the question of internal mechanism of what makes adaptive/non-adaptive networks different, and how training pushes networks towards one or the other, along with pointers to possible explanations.

My other issues were appropriately addressed, and I was happy to see the addition of another task and presentation of hyperparameter robustness. Thanks again for the nice presentation and effort into language and figures.

Point by point responses:

> "In the original paper, we partially explained this by noting that networks that learned multiple movements create intermediate states that can be used during adaptation. We supported this by showing that these networks can produce intermediate movements when prompted with intermediate inputs that they have not explicitly learned (Fig. 2i, Fig. S5b). Also, when we applied noise to the activity when networks were producing the first movement, the output moved towards the output of the second movement, suggesting that there was interpolation between the movements in both neural space and motor output (Fig. 2g). To make a more direct connection between the neural space and motor output, we have now plotted both the latent trajectories and motor output when noise was applied to the activity (Fig. R3a,b,d,e). Indeed, noise pushed networks to intermediate states in neural space that corresponded to intermediate motor output. That networks were able to access these intermediate states with noise added suggests that these states are close to the states associated with the original movement.

This noise analysis may answer other questions, but it does not by itself demonstrate the mechanism by which multi-movement networks develop representations that facilitate adaptation. The authors are hypothesizing that an important feature for adaptation is nearness in representational space (by a Euclidean metric) to states that allow output remapping. This sounds like a reasonable hypothesis. Can this be tested? Such a test would help to make it explicit in the main text that this is the central hypothesis about what about internal states enable fast remapping.

> "Regarding analytical derivations, while they are indeed very powerful and elegant, they become intractable for recurrent networks like the ones we have modeled, which is why most of the

current work in the references above focus on feedforward networks.”

This is true – in RNNs this analysis is difficult. I would still find it satisfying to see analyses in feedforward nets, as it’s likely that many of the same phenomena appear in feedforward nets. If this is left for future work, a discussion paragraph about the mystery of mechanism would be nice. Such a paragraph should certainly mention that in simplified settings (e.g. linear readouts from states, i.e. kernel machines), there exists a clean mathematical understanding of when things are easily learnable, and these may give insight into present observations.

Reviewer #3 (Remarks to the Author):

The authors have done an outstanding job with the revisions. This paper will be a valuable contribution to the literature, and I look forward to seeing it out.

Response to reviewer comments

Reviewer #1 (Remarks to the Author):

I thank the authors for their very thorough revisions which, in my opinion, have very significantly strengthened the flow of the arguments and overall strength of their claims. The revisions have largely addressed my main concerns with one exception:

In my initial review I highlighted that the link between what their model is testing and the concept of "de novo learning" as described in the motor learning literature, is not necessarily a straight forward one. Since it's not completely straight forward, I don't fundamentally object to them retaining the framing within the manuscript but think the manuscript shouldn't be published without including a discussion of these nuances and potential limits on how their model maps to experiments and existing literature/concepts around de novo learning.

We thank the reviewer for their enthusiasm about our revisions. We have included a discussion about *de novo* learning in the text.

Reviewer #2 (Remarks to the Author):

Thank you for the comprehensive responses. The changes made to the manuscript have addressed some of my concerns, but not all. In particular, I continue to find it unsatisfying that it is not analyzed why (or mechanistically how) these neural network models find it easier to adapt after certain training regimens. In my opinion the time has passed where it is sufficient to present a simple comparison of brains and ANNs as an explanation of a brain phenomenon. It is time, and indeed very possible, to look deeper into the reasons why these phenomena occur.

Seeing as it is difficult to use theory to answer this question, it would still be satisfying to see analyses testing hypotheses about how, mechanistically speaking, the repertoire affects future learning. At a minimum, I would like to see a discussion paragraph asking the question of internal mechanism of what makes adaptive/non-adaptive networks different, and how training pushes networks towards one or the other, along with pointers to possible explanations.

We thank the reviewer for their suggestions. Our comparison and "dissection" of the adaptation processes following exposure to the same perturbation across networks trained on different movement repertoires through different learning experiences was aimed at understanding how these factors shape neural population structure and subsequent learning. We have included additional text in the discussion addressing the question of internal mechanism focusing on ideas from the computation through dynamics framework (e.g., Vyas et al *Annu Rev Neurosci* 2020).

My other issues were appropriately addressed, and I was happy to see the addition of another task and presentation of hyperparameter robustness. Thanks again for the nice presentation and effort into language and figures.

We thank the reviewer for their enthusiasm about our revisions.

Point by point responses:

> “In the original paper, we partially explained this by noting that networks that learned multiple movements create intermediate states that can be used during adaptation. We supported this by showing that these networks can produce intermediate movements when prompted with intermediate inputs that they have not explicitly learned (Fig. 2i, Fig. S5b). Also, when we applied noise to the activity when networks were producing the first movement, the output moved towards the output of the second movement, suggesting that there was interpolation between the movements in both neural space and motor output (Fig. 2g). To make a more direct connection between the neural space and motor output, we have now plotted both the latent trajectories and motor output when noise was applied to the activity (Fig. R3a,b,d,e). Indeed, noise pushed networks to intermediate states in neural space that corresponded to intermediate motor output. That networks were able to access these intermediate states with noise added suggests that these states are close to the states associated with the original movement. This noise analysis may answer other questions, but it does not by itself demonstrate the mechanism by which multi-movement networks develop representations that facilitate adaptation. The authors are hypothesizing that an important feature for adaptation is nearness in representational space (by a Euclidean metric) to states that allow output remapping. This sounds like a reasonable hypothesis. Can this be tested? Such a test would help to make it explicit in the main text that this is the central hypothesis about what about internal states enable fast remapping.

We were also interested in finding compact geometrical principles that fully predicted adaptation patterns, but after careful exploration we think that due to the multiple interactions that shape this process—existing structure in neural activity space, type of inputs, nature of outputs, changes imposed by the perturbation—, there may not be a very mathematically tractable relationship. Nonetheless, we remain interested in how the available structure of neural activity shapes the activity and network changes necessary for future learning, and in fact are exploring this topic in a follow up study where we are modeling Brain-Computer Interface adaptation tasks. We hope that this follow up work will provide more direct insights given the nature of the tasks and perturbations that we are modeling.

> “Regarding analytical derivations, while they are indeed very powerful and elegant, they become intractable for recurrent networks like the ones we have modeled, which is why most of the current work in the references above focus on feedforward networks.”

This is true – in RNNs this analysis is difficult. I would still find it satisfying to see analyses in feedforward nets, as it’s likely that many of the same phenomena appear in feedforward nets.

If this is left for future work, a discussion paragraph about the mystery of mechanism would be nice. Such a paragraph should certainly mention that in simplified settings (e.g. linear readouts from states, i.e. kernel machines), there exists a clean mathematical understanding of when things are easily learnable, and these may give insight into present observations.

We thank the reviewer for their suggestions. We have included additional text in the discussion addressing the question of internal mechanism.

Reviewer #3 (Remarks to the Author):

The authors have done an outstanding job with the revisions. This paper will be a valuable contribution to the literature, and I look forward to seeing it out.

We thank the reviewer for their enthusiasm for our work, and for all their suggestions that improved our manuscript.